 

# Integrated analysis of microbiome and metabolome reveals signatures in PDAC tumorigenesis and prognosis

Yuan Fang,[1,2,3,4] Xiaohong Liu,[1,2,3,4] Jie Ren,[1,2,3,4] Xing Wang,[1,2,3,4] Feihan Zhou,[1,2,3,4] Shi Huang,[5] Lei You,[1,2,3,4] Yupei Zhao[1,2,3,4]

**ABSTRACT** Pancreatic cancer, predominantly pancreatic ductal adenocarcinoma (PDAC), is one of the most malignant tumors of the digestive system. Emerging evidence suggests the involvement of the microbiome and metabolic substances in the development of PDAC, yet the results remain contradictory. This study aims to identify the alterations and relationships in intratumoral microbiome and metabolites in PDAC. We collected matched tumor and normal adjacent tissue (NAT) samples from 105 PDAC patients and performed a 6-year follow-up. 2bRAD-M sequencing, untargeted liquid chromatography-tandem mass spectrometry, and untargeted gas chromatography-mass spectrometry were performed. Compared with NATs, microbial α-diversity decreased in PDAC tumors. The relative abundance of *Staphylococcus aureus*, *Cutibacterium acnes,* and *Cutibacterium granulosum* was higher in PDAC tumor after adjusting for confounding factors body mass index and M stage, and the presence of *Ralstonia pickettii_B* was found associated with a worse overall survival. Metabolomic analysis revealed distinctive differences in composition between PDAC and NAT, with 553 discriminative metabolites identified. Differential metabolites were revealed to originate from the microbiota and showed significant interactions with shifted bacterial species through KO (KEGG Orthology) genes. These findings suggest that the PDAC microenvironment harbors unique microbial-derived enzymatic reactions, potentially influencing the occurrence and development of PDAC by modulating the levels of glycerol-3-phosphate, succinate, carbonate, and beta-alanine.

**IMPORTANCE** We conducted a large sample-size pancreatic adenocarcinoma microbiome study using a novel microbiome sequencing method and two metabolomic assays. Two significant outcomes of our analysis are: (i) commensal opportunistic pathogens *Staphylococcus aureus*, *Cutibacterium acnes*, and *Cutibacterium granulosum* were enriched in pancreatic ductal adenocarcinoma (PDAC) tumors compared with normal adjacent tissues, and (ii) worse overall survival was found related to the presence of *Ralstonia pickettii_B*. Microbial species affect the tumorigenesis, metastasis, and prognosis of PDAC via unique microbe-enzyme-metabolite interaction. Thus, our study highlights the need for further investigation of the potential associations between pancreatic microbiota-derived omics signatures, which may drive the clinical transformation of microbiome-derived strategies toward therapy-targeted bacteria.

**KEYWORDS** PDAC, pancreatic ductal adenocarcinoma, microbial metabolism, microbiota, metabolome, carcinogenesis

P ancreatic ductal adenocarcinoma (PDAC) remains a highly fatal malignancy, with a 5-year overall survival (OS) of 13% (1–4). Therapeutic methods for PDAC remain limited (5). Surgery combined with adjuvant chemotherapy remains the only curative therapeutic option, but more than 60% of patients are diagnosed with unresectable disease (6). The pathogenesis of PDAC is complex, and genetic alterations in PDAC

Address correspondence to Lei You, florayo@163.com, or Yupei Zhao, zhao8028@263.net.

The authors declare no conflict of interest.

See the funding table on p. 15.

fail to explain carcinogenesis alone, which leaves environmental factors, including the microbiota, emerging as potential mediators of PDAC carcinogenesis. Thus, as an essential hallmark of cancer (7), the role of polymorphic microbiomes in cancer remains to be explored, and there is a pressing need to identify microorganisms that might explain the differences between PDAC tumors and normal pancreatic tissue so that new concepts can be developed for future therapies (8).

Numerous studies have shown that microorganisms are critical in carcinogenesis (9, 10). Intratumoral bacteria have been observed in various tumors, including PDAC (11), which is associated with PDAC carcinogenesis, progression, and poor prognosis via a complex mechanism. Geller et al. initially proposed a connection between intratumoral microbiota and PDAC (12). The prevalence of intratumoral bacteria in PDAC tissues was significantly higher than in normal pancreatic tissues. The presence of microbiota in the pancreas of both healthy and cancerous subjects was also confirmed in a study by Thomas et al. (13). In a study involving 12 PDAC patients, Pushalka et al. found higher bacterial biomass in PDAC tumors than in normal pancreatic tissue (14, 15). α-Diversity was reported to be slightly higher in healthy controls versus in patients (16). The most common class identified in the PDAC intratumor microbiome is Gammaproteobacteria, with the dominant genus *Pseudomonas* (12, 14), which carries long-form cytidine deaminase that metabolizes the chemotherapeutic drug gemcitabine (2′,2′-difluorodeoxycytidine) into its inactive form (2′,2′-difluorodeoxyuridine) (12). Riquelme et al. investigated the impact of tumor microbiota on PDAC patient survival (17). Patients with long-term survival exhibited higher α-diversity and enrichment for *Pseudoxanthomonas*, *Saccharopolyspora*, and *Streptomyces*. Guo et al. revealed that *Acinetobacter*, *Pseudomonas*, and *Sphingopyxis*, intratumoral microbiota of basal-like PDAC, were associated with worse prognosis by inducing inflammation (18). These studies support that distinctive profiles of tumor microbiota may underlie PDAC heterogeneity, and comprehensive characterization of the PDAC intratumoral microbiome may be an essential step in unraveling the effects of bacteria on PDAC tumorigenesis and prognosis. Despite these developments, the clinical significance of the intratumoral microbiome in PDAC is still poorly understood. Previous comparative PDAC-healthy control studies were generally constrained by the limitations of a small number of samples and the vague classification of taxonomy. Thus, extensive sample-size studies are urgently required. Therefore, further investigation of the PDAC tumor microbiome's profiles and clarifying its clinical significance and prognostic value re imperative.

Microbiota-derived metabolites are important natural products that establish a strong connection between the microbiome and cancer (19). For example, microbial byproducts can actively contribute to carcinogenesis. Secondary metabolites, including lithocholic acid and deoxycholic acid (20, 21), as well as catabolites, such as acetate and butyrate (22, 23), play a crucial role in enhancing either epithelial-mesenchymal transition or cell proliferation in several models of cancer (24). Metabolomic comparisons of human PDAC tumor tissue and normal adjacent tissue (NAT) revealed that tumor tissues exhibit lower levels of glucose, upper glycolytic intermediates, creatine phosphate, and the amino acids glutamine and serine, which are the primary metabolic substrates (25). However, evidence of the involvement of microbiome-derived metabolites in PDAC carcinogenesis is limited.

In summary, to advance our understanding of the microbiome and metabolome characteristics associated with PDAC and to elucidate the intricate role of their interaction in PDAC carcinogenesis and prognosis, we conducted comprehensive analyses of the microbiome and metabolome of surgically excised PDAC tumor and its matched NATs from a large scale of 105 patients based on 2bRAD-M sequencing, untargeted liquid chromatography-tandem mass spectrometry (LC-MS), and untargeted gas chromatography-mass spectrometry (GC-MS). Our study provided data support for subsequent studies on PDAC, thereby expanding the perspective in this field.

## RESULTS

### Participant characteristics

We collected 208 tissue samples (103 matched PDAC and NATs, plus unpaired 2 NATs) from 105 patients. The demographic, clinical, and pathological characteristics of the patients are shown in Table 1. The average age was 61.57 ± 8.18, and 63 patients (60.0%) were male. Diabetes was present in 39 (37.1%) patients, and among them, 14 (13.3%) were new-onset diabetes. The number of patients with tumor size, node, and metastasis (TNM) stages I, II, III, and IV were 36 (34.3%), 47 (32.4%), 19 (18.1%), and 3 (2.9%), respectively. Tumor differentiation was well (16.1%), moderate (41.0%), and poor (42.9%).

### PDAC and NAT microbiome differ from global scale

Negative controls were collected in the operating room and laboratory to remove the interference of contaminating microorganisms introduced during the sample collection and experimental manipulation. Using a combination of decontam, microDecon, and FEAST, background microorganisms were deducted based on the negative controls. After decontamination, 1,920 species were identified. A broad overview of our taxonomic data from the 105 subjects is provided in Fig. S1.

α-Diversity was calculated at the species level to compare differences between groups. Significant decreases in Chao1, Shannon, and Simpson index were observed in PDAC (Fig. 1A; $P = 0.0017$, $0.00012$, and $0.00089$, respectively). Due to the significant differences in α-diversity between NAT and PDAC tissues, we further examined the compositional diversity of the microbiota in these two groups. A Venn diagram of the microbiota composition revealed that NAT had a greater variety of microorganisms (Fig. 1B). At the species level, 512 microorganisms were shared by the NAT and PDAC groups. Principal coordinate analysis (PCoA) based on the Bray-Curtis distance was performed on the samples (Fig. 1C). The results of the permutational multivariate analysis of variance (PERMANOVA) test indicated a statistically significant difference in the β-diversity between PDAC and NAT ($R^2 = 0.011$; $P = 0.01$).

Next, we investigated the potentially relevant influence factors for microbiome alterations. PERMANOVA was used to explore the associations between variations in the pancreatic microbiota and host characteristics. Given a false discovery rate (FDR) of 5%, three parameters were significantly associated with microbial variations derived from Bray-Curtis distances calculated on the species level (Fig. 1C). Group, M stage, and body mass index (BMI) level were explanatory factors consistent with previous research.

### Taxonomic signatures of microbiota in PDAC tumor

Next, we attempted to identify PDAC-associated taxa using multivariable microbiome associations with a linear model (MaAsLin 2) to control for confounding factors (26). The model included group as the fixed effect and BMI and M stage as random effects. Thirteen species were identified as differentially abundant bacterial species between PDAC and NAT (Fig. 2A and C, Table S1). Samples from the PDAC group had higher levels of *Staphylococcus aureus*, *Cutibacterium acnes*, and *Cutibacterium granulosum*, while having lower levels of *Sphingomonas aquatilis*, *BACL27 sp014190055*, *QWOQ01 sp003669585*, *Limnohabitans_A sp005789685*, *Mycobacterium koreense*, *Mycobacterium intermedium*, *UBA953 sp002293125*, *Bacillus_A bombysepticus*, *Pelomonas sp003963075*, and *Dialister hominis* (Table S2). After adjusting for the M stage, the associations were still significant. To compare our results with those of previous studies, we performed MaAsLin 2 at the genus level, and the results were broadly consistent with the species level. Besides, *Bifidobacterium* was slightly increased in PDAC, while *Dietzia* and *Streptococcus* were depleted (Fig. 2B and D).

According to the PERMANOVA results (Fig. 1C), we conducted an analysis of bacterial changes associated with the M stage. The pathogens *Pseudomonas fulva*, *Dietzia maris*, *Massilia timonae,* and *Brevundimonas diminuta* were positively associated with the M1 stage, whereas *Pseudomonas_E sp900187635* was depleted in the M1 stage (Fig. S3). In a

**TABLE 1** Characteristics of PDAC patients enrolled

| Characteristics | Count |
|---|---|
| $n$ (Patient) | 105 |
| Gender = male (%) | 63 (60.0) |
| Age [mean (SD)] | 61.57 (8.18) |
| Family history = yes (%) | 20 (19.0) |
| Pancreatitis = yes (%) | 31 (30.1) |
| Other malignancy = yes (%) | 8 (7.6) |
| BPD[a] = yes (%) | 17 (16.2) |
| EUS FNA[b] = yes (%) | 11 (10.5) |
| Weight loss [mean (SD)] | 4.19 (4.82) |
| Body mass index [mean (SD)] | 24.76 (3.20) |
| Smoking (%) | |
| Never | 58 (55.2) |
| Ever | 12 (11.4) |
| Current | 35 (33.3) |
| Diabetes (%) | |
| New-onset | 14 (13.3) |
| No | 66 (62.9) |
| Yes | 25 (23.8) |
| Alcohol (%) | |
| Never | 65 (61.9) |
| Ever | 2 (1.9) |
| Current | 38 (36.2) |
| Hyperlipidemia (%) | |
| Dyslipidemia | 14 (13.3) |
| No | 58 (55.2) |
| Yes | 33 (31.4) |
| Biliary disease = yes (%) | 62 (59.0) |
| Antibiotics = yes (%) | 10 (9.5) |
| Location = tail (%) | 44 (41.9) |
| CA19-9 upregulate = yes (%) | 85 (81.0) |
| Differentiation (%) | |
| Poor | 45 (42.9) |
| Moderate | 43(41.0) |
| Well | 17 (16.1) |
| Perineural invasion = yes (%) | 74 (70.5) |
| Blood vessel invasion = yes (%) | 40 (38.1) |
| T stage (%) | |
| T1 | 27 (25.7) |
| T2 | 47 (44.8) |
| T3 | 28 (26.7) |
| T4 | 3 (2.9) |
| N stage (%) | |
| N0 | 51 (48.6) |
| N1 | 37 (35.2) |
| N2 | 17 (16.2) |
| M stage = M1 (%) | 3 (2.9) |
| Stage (%) | |
| IA | 15 (14.3) |
| IB | 21 (20.0) |
| IIA | 13 (12.4) |
| IIB | 34 (32.4) |
| III | 19 (18.1) |

**TABLE 1** Characteristics of PDAC patients enrolled (*Continued*)

| Characteristics | Count |
|---|---|
| IV | 3 (2.9) |
| Neoadjuvant therapy = yes (%) | 4 (3.8) |

[a]BPD, Bbiliary and pancreatic duct drainage
[b]EUS-FNA, Eendoscopic ultrasound-guided fine-needle aspiration.

subset containing only PDAC samples, these results remained consistent. However, it is important to note that only three patients were diagnosed with stage M1, rendering this result incidental.

To investigate the function of the intratumoral microbiota in PDAC, we predicted the biological functions of the bacteria utilizing PICRUSt2. We identified 7,301 KO (KEGG Orthology) genes altogether. Using the MaAsLin 2 analysis, 1,079 KO genes were found to be differentially expressed between the two groups after adjusting for BMI (Supplementary data).

## Microbial species related to overall survival

Analysis of the relationship between microbial species and overall survival is performed only in PDAC tumor samples. Among them, one patient with perioperative cardiac death and four patients with uncertain time of death were excluded. A total of 98 PDAC patients were included in the survival analysis, resulting in a median of 15 months of follow-up (range 1–71 months).

In 100 times 10-fold cross-validated elastic-net Cox regression models for OS, we found species *Ralstonia pickettii_B* and age were selected >50% of the time with $P < 0.20$ in the standard univariate Cox regression. Based on these results, we then tested the relationship between *Ralstonia pickettii_B* and OS by stratifying the patients in two groups based on the presence of *Ralstonia pickettii_B*. As expected, we found that patients colonized with *Ralstonia pickettii_B* had significantly worse OS (median OS: 17 months) than those *Ralstonia pickettii_B* negative ones (median OS: 37 months) using Kaplan-Meier curve tested by log-rank [hazard ratio (HR), 2.79; 95% CI, 0.98–7.94; $P = 0.045$] (Fig. 3A; Table S2). Given that PDAC is a disease in which risk increases with age, we stratified the patients into two groups by age 65. A median OS of 36 months was obtained for middle-aged group and 17 months for elder patients group (HR, 2.10; 95% CI, 1.06–4.19; $P = 0.0047$) (Fig. 3B). Subgroup analyses of age and colonization of *Ralstonia pickettii_B* found that only the *Ralstonia pickettii_B*-negative group had a difference in OS between middle-aged and older adults, but the older group was too under-represented. Our follow-up is ongoing, and the existing results will be updated as the study continues. Our findings indicate that the presence of *Ralstonia pickettii_B* in the tumor could predict survival outcome in resected PDAC patients, further elaborating the potential of the microbiome composition in mediating PDAC progression.

## Untargeted metabolomics profiling revealed significantly altered metabolites

Untargeted metabolomic profiling was performed on a subset of 98 NAT and 90 PDAC samples to investigate the interactions between the pancreatic microbiota and host-microbe co-metabolism.We conducted LC-MS and GC-MS to make our assay as comprehensive as possible. A total of 6,375 metabolites were quantified from tissue samples using LC-MS and 481 were quantified using GC-MS.

Orthogonal partial least squares-discriminant analysis (OPLS-DA) (Fig. 4A and B) showed differences in the tissue metabolite profiles between PDAC and NAT groups, indicating a tumor-metabolite shift in PDAC carcinogenesis. The ability of the OPLS-DA model was tested during a seven cross-validation through 200 random permutation tests. The intercepts of goodness-of-fit ($R^2$) and goodness-of-prediction (Q2) illustrate that the OPLS-DA model is reliable and does not overfit. We plotted fold changes using volcano plots of the levels of identified metabolites in PDAC relative to NAT samples,

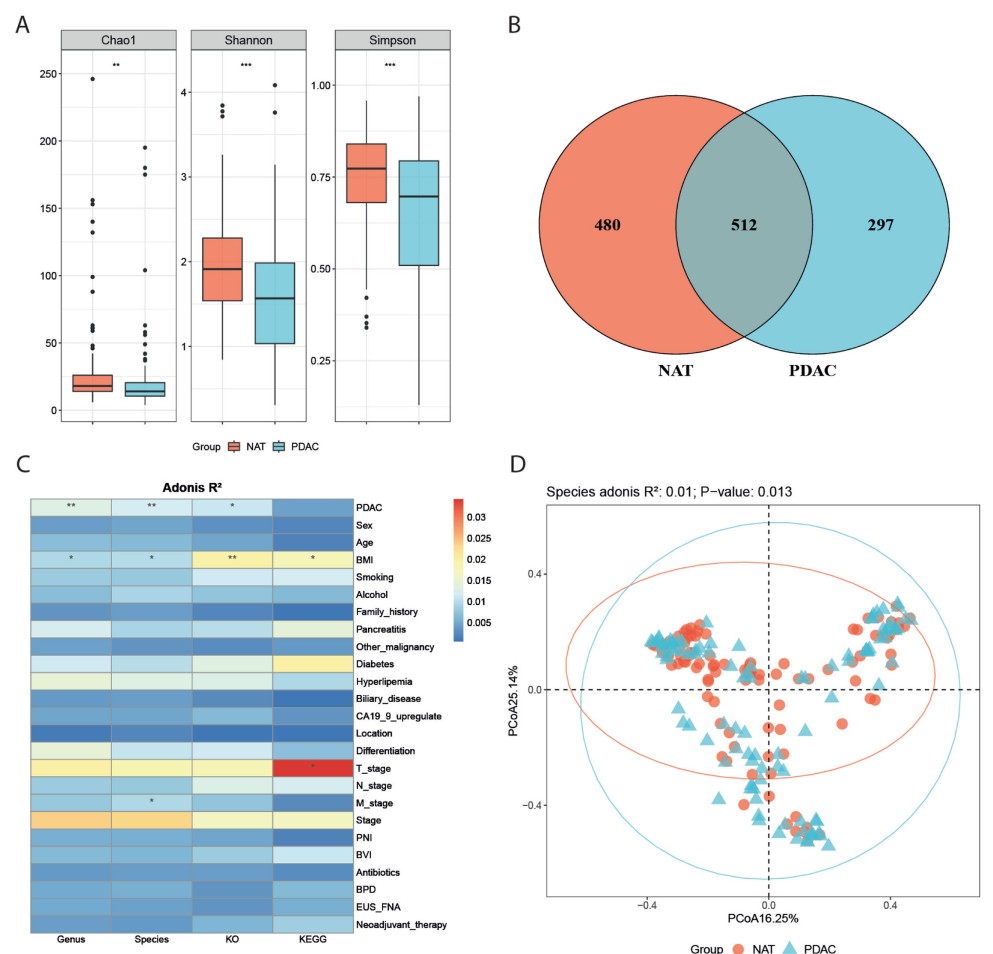

**FIG 1** Pancreatic microbiota dysbiosis in PDAC. (A) α-Diversity between PDAC and NAT base on Chao1, Shannon, and Simpson indices ($P = 0.0017$, 0.00012, 0.00089). The box represented the interquartile range (IQR) between the first and the third quartiles, and the midline represented the median. (B) Venn diagram shows numbers of species observed in PDAC and NAT. (C) Confounder factor analysis using PERMANOVA test based on Bray-Curtis distance with 999 permutations, *$P <$ 0.05 and **$P <$ 0.01. (D) PCoA for PDAC (blue) and NAT samples (pink) based on Bray-Curtis distance ($R^2 = 0.011$; $P = 0.01$).

considering the statistically significant difference (*P*-value) and variable importance in the projection. As shown in Fig. 4D and E, the levels of the differential metabolites in PDAC were significantly different from those in NAT in LC-MS and GC-MS profiling. PDAC was associated with significant changes in the metabolome from LC-MS and GC-MS profiling.

## The altered metabolites and KEGG pathways in PDAC tissues compared with NAT

We then investigated the association of each annotated metabolite with the PDAC group. We identified 417 different metabolites in PDAC tissues compared with NAT, including 138 elevated and 279 depleted metabolites from LC-MS profiling [variable importance in projection (VIP) > 1 and *Q* value <0.05] (Fig. 4D). We identified 147 differential metabolites using GC-MS profiling, of which 17 were elevated and 130 were depleted (Fig. 4E). Metabolite origin analysis using MetOrigin revealed 149 differential metabolites associated with host and microbiota, including 4 host-specific metabolites, 43 bacterial metabolites, and 102 bacteria-host co-metabolites (Fig. 4C). The abundance of 145 bacterial-related metabolites was shown as a heatmap (Fig. S5). The depleted metabolites glycerophosphocholine and 2-lysophosphatidylcholine had the highest VIP

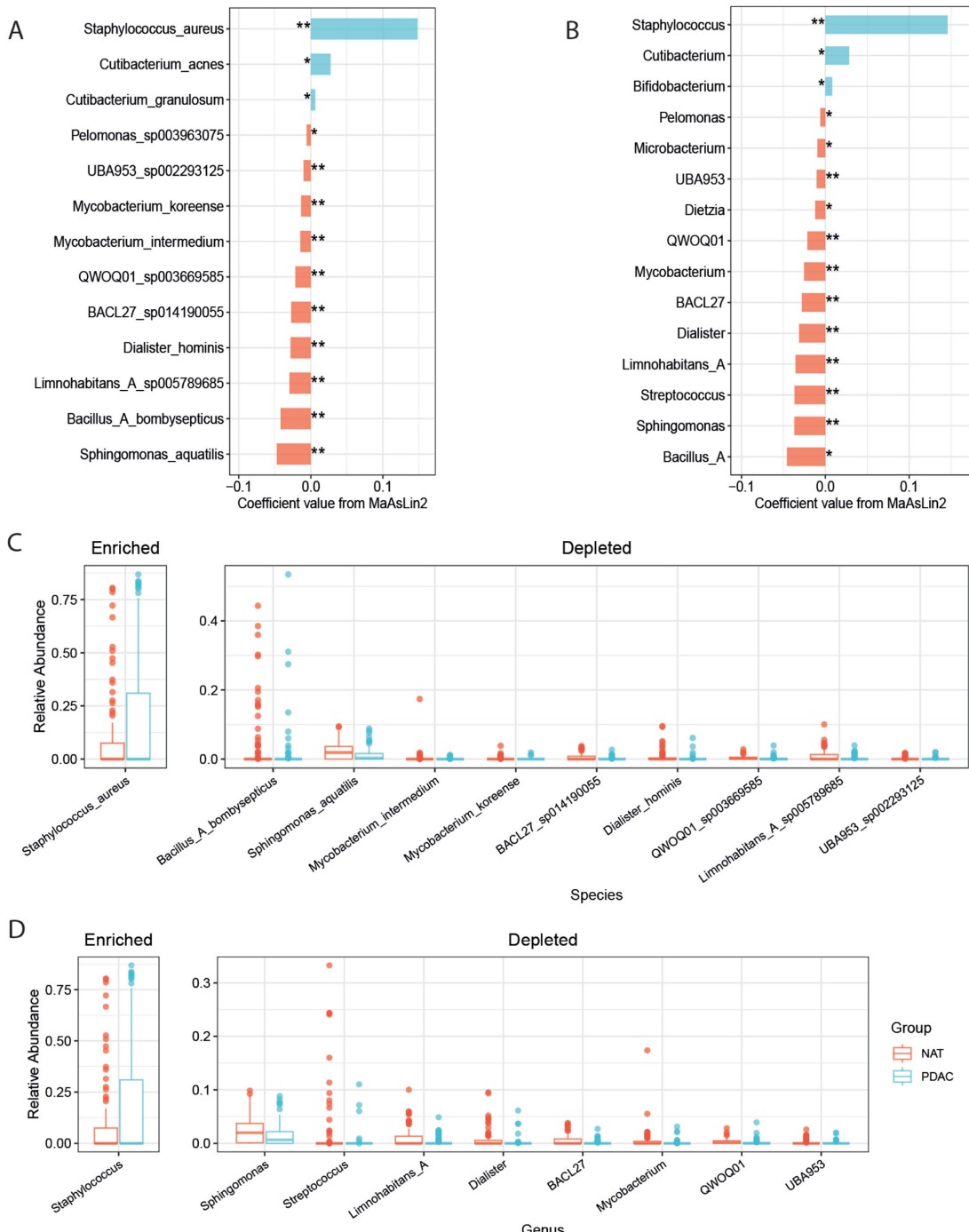

**FIG 2** Differential relative abundance of pancreatic microbiota in PDAC and NAT. (A and B) Differential taxa at the species and genus level identified by microbiome multivariable associations with linear model (MaAsLin 2) adjusted for confounding factors BMI and M stage (*q < 0.25) .** represent LEfSe analysis significant (LDA >2). (C and D) Boxplot showed relative abundance of group-distinct bacterial species and genus with LEfSe analysis significant.

scores, whereas the most important enriched metabolites were oleamide, palmitoylcar-nitine, and L-acetylcarnitine. Among the amino acid metabolites, beta-alanine, ergothio-neine, and L-isoleucine levels were significantly increased in PDAC, whereas other amino acids and analogs, such as L-isoleucine, L-valine, L-aspartic acid, L-cysteine, L-serine, and L-glutamine were significantly reduced in PDAC samples. The roles of these altered

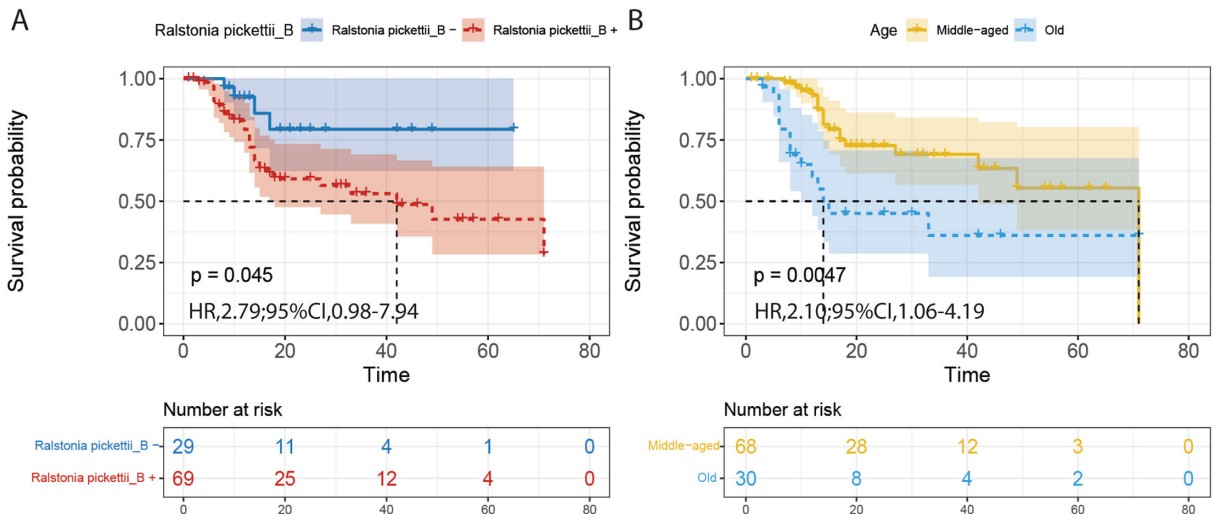

**FIG 3** Prognostic microbial biomarker. Kaplan-Meier curves illustrating the difference in overall survival in tumor samples stratified by (A) presence of microbial species *Ralstonia pickettii_B*. and (B) age. *P*-values were calculated using log-rank test. HRs and 95% CIs were calculated using univariate Cox regression analysis.

metabolites in PDAC need to be further studied, allowing for potential correlation analysis based on metabolite-microbial interactions.

We conducted metabolite pathway enrichment analysis (MPEA) on differential metabolites from the host, microbiota, and bacteria-host co-metabolites. There were 3 and 82 metabolic pathways related to the bacteria and co-metabolism pathway, respectively (Fig. 4F). Among these, 3 and 32 metabolic pathways were identified as significantly associated with PDAC correspondingly (*P* < 0.05) (Fig. 4G). Based on origin-based function analysis, no metabolic pathway was found specifically related to host, while ascorbate and aldarate metabolism and folate biosynthesis were specific to the bacteria alone, and 32 metabolic pathways associated with amino acids, lipids, and sugars were shared by both host and microbiota.

## The association between discriminative species and metabolites

Our multi-omics data enabled us to identify dynamic interactions between differential taxonomic and metabolic signatures. To dissect interactions between the host and microbiota that might underlie features in PDAC, we assessed the correlations between PDAC-related species, altered KO genes, and differentially abundant metabolites originating from microbiota in PDAC and NAT, respectively. Furthermore, to explore more accurate evidence of microbial enzyme-metabolite interactions, based on the reactions in the KEGG (Kyoto Encyclopedia of Genes and Genomes) database, we associated the altered metabolites with the discriminate KO genes, which significantly correlated to both species and metabolites.

Broadly, in the Spearman correlation analysis, we observed more significant correlations in the NAT samples (Fig. S6A and B). A total of 46 differential metabolites were found to be significantly correlated with eight differential species, and 302 KO genes were identified to be concurrently associated with both, resulting in a cumulative total of 2,523 correlations (Fig. S6C). Among them, the PDAC-enriched species *Cutibacterium acnes* and *Cutibacterium granulosum* were broadly negatively correlated with differentially abundant metabolites in NAT samples (Fig. S6A). In PDAC samples, 397 KO genes were found to form 1,260 correlations with 16 differential metabolites and five differential species (Fig. S6D). *Cutibacterium granulosum* was only negatively correlated with D-ornithine, and *Cutibacterium acnes* was only negatively correlated with EPA (d5) (eicosapentaenoic acid). Meanwhile, *Cutibacterium acnes* and *Cutibacterium granulosum*

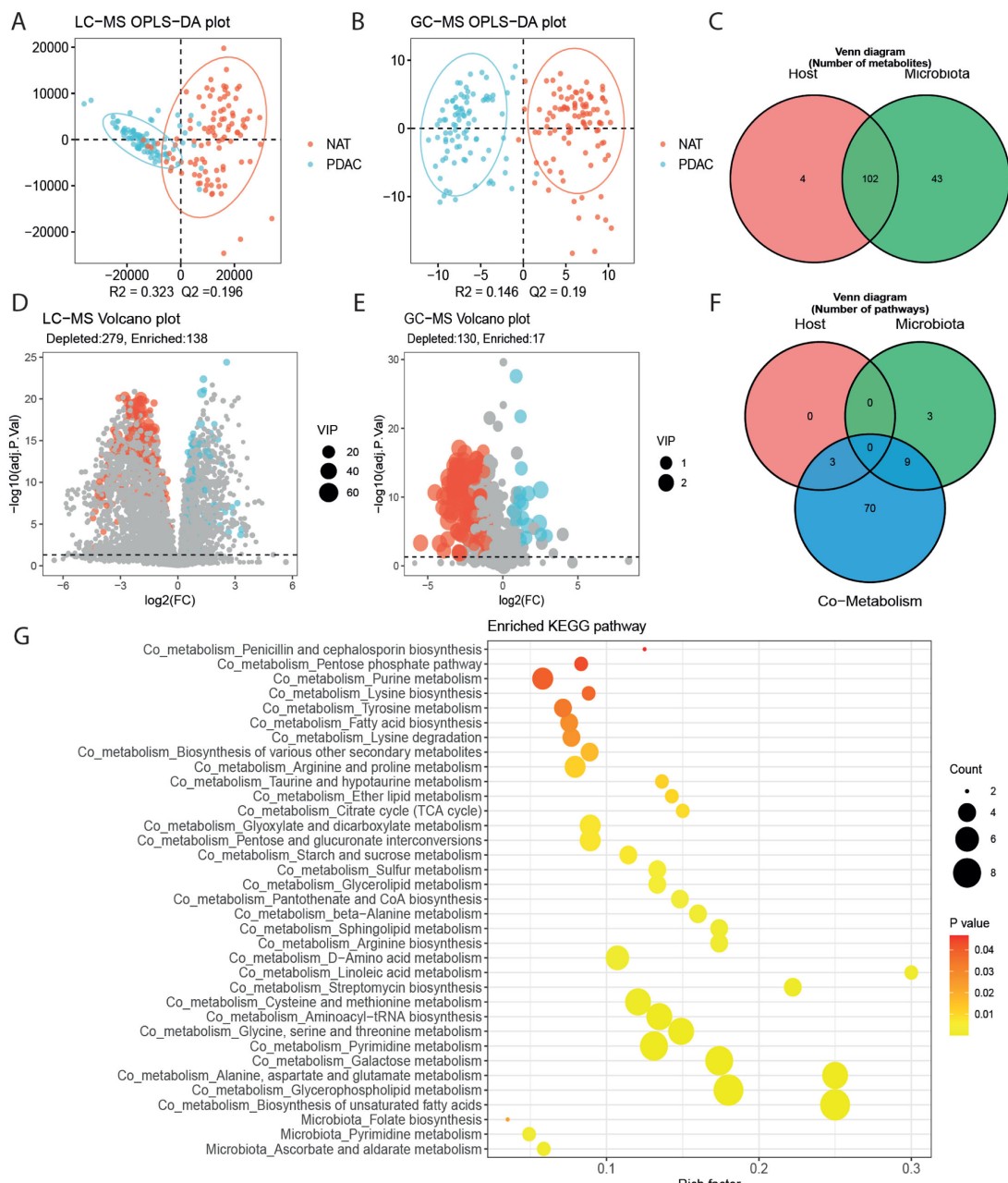

**FIG 4** Metabolome profile and function changes in PDAC (A) and (B) OPLS-DA showed that PDAC and NAT samples were separated into two distinct clusters. (D, E) Volcano plot demonstrated metabolite changes between 90 PDAC and 98 NAT samples. The x-axis indicates log2-transformed fold change of metabolite abundances, and the y-axis denotes log10-transformed *Q* values (*P*-value adjusted using the tail area-based FDR). The horizontal lines represent *q* < 0.05. (C and F) Venn plot showed results of metabolites origin analysis. (G) The functions of discriminative metabolites derived from microbiota were analyzed using the KEGG (Kyoto Encyclopedia of Genes and Genomes) database, and the enriched metabolic pathways are presented in a bubble plot. The size of bubble represents the number of metabolites detected in the KEGG pathway.

formed new positive correlations with carbonate, and *Cutibacterium acnes* formed a new negative correlation with L-serine (Fig. S6B).

In the representative formula listed for chemical reactions involving differential metabolite and bacterial enzyme genes (Fig. 5C and D; Table S3), the levels of substrate sn-glycero-3-phosphoethanolamine and its product glycerol-3-phosphate (G3P) were significantly reduced in PDAC samples, while the enzymes metabolizing them, namely glpQ and ugpQ, were elevated. Similarly, pcrB, which catalyzes the conversion

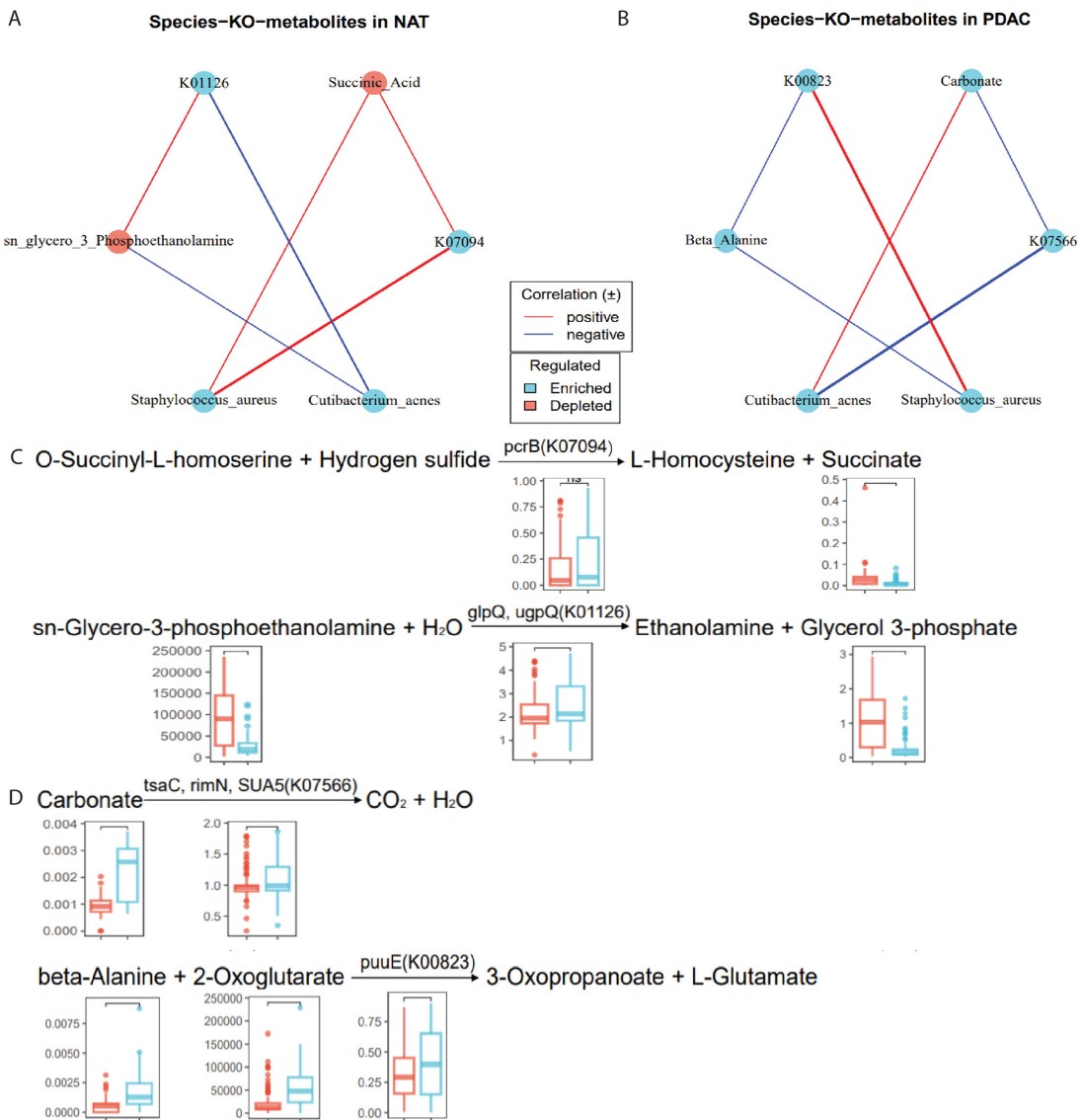

**FIG 5** Integrated analysis of multi-omics in NAT and PDAC. (A, B) The network revealed representatively significant and suggestive associations ($P < 0.05$, | r | > 0.3, Spearman analysis) among differentially abundant taxa, metabolites, and KO genes, (A) NAT group, and (B) PDAC group. Lines connecting nodes indicate positive (red) or negative (blue) correlations. (C, D) Representative metabolites-KO gene reactions. Representative enzymes, metabolites appearing in the existing PDAC metabolite-enzyme reactions are shown in the formula listed. Each boxplot in a reaction represents a compound or a KO gene (two-side Wilcoxon rank-sum test). Boxplots display the average abundance of metabolites or KO genes. (C) NAT group and (D) PDAC group.

of O-succinyl-L-homoserine and hydrogen sulfide to L-homocysteine and succinate, respectively, was upregulated in PDAC samples. In contrast, the levels of its reaction product succinate decreased. In other identified chemical reactions, enzymes tsaC, rimN, and SUA5, responsible for carbonate decomposition, were significantly enriched in carbonate content. Additionally, beta-alanine and 2-oxoglutarate were significantly enriched compared to adjacent control tissues, and the enzyme gene puuE catalyzing their reaction to produce glutamate was also significantly enriched. In summary, though it remains further explored, our analysis indicates that the PDAC microenvironment exhibits distinct enzymatic reactions and metabolic processes originating from microorganisms.

## DISCUSSION

This study reports an integrated analysis of the pancreatic microbiome, metabolome, and predicted microbial KO genes in patients with PDAC. Owing to the unavailability of healthy pancreatic specimens, we used matched normal adjacent tissue specimens; comparison between matched PDAC and NAT minimized interpatient confounding factors such as diet or lifestyle, which are already known to impact commensal microbiome composition significantly (27, 28). Integrated analysis showed unique reactions in PDAC, providing new mechanistic insights into the pathogenesis of the disease. In this study, the α-diversity of pancreatic microbiota was higher in normal adjacent tissues than in tumor tissues, consistent with the results of a previous study (16, 29). Many altered differential microbial species and metabolites have been identified between PDAC and NAT samples, which may indicate a general mechanism for PDAC.

The colonization of bacteria in PDAC has been demonstrated by several studies. In our study, the composition of the high-abundance intratumoral bacteria observed was consistent with previous PDAC studies (29, 30). The dominant phyla identified were Proteobacteria and Firmicutes, with Gammaproteobacteria emerging as the most dominant microbial class. At the genus level, the dominant genera identified in our study, such as *Pseudomonas*, *Staphylococcus*, *Ralstonia*, and *Sphingomonas*, have also been recognized as dominant taxa in previous studies (29, 30). However, our findings on the differential microbial species between PDAC and NAT differ from previous studies, which also reported great variability (29–31). Specifically, our research identified that commensal opportunistic pathogens, including *Staphylococcus aureus*, *Cutibacterium acnes*, and *Cutibacterium granulosum*, were enriched in PDAC tumor. Although these findings differ from earlier pancreatic cancer studies, they aligned with alterations observed in other tumors, indicating the potential roles in tumorigenesis (32, 33). Cavarretta et al. described increased *Staphylococcus aureus* and *Cutibacterium acnes* in prostate cancer. *Staphylococcus aureus* was found to colonize the tumor tissue of breast cancer patients, and the intratumor *Staphylococcus* significantly contributed to tumor metastasis in animal experiments (33). Inflammation promotes the development of tumors (34, 35). *Cutibacterium acnes* has phosphatidylinositol and peptidoglycan in the outer envelope, which contribute to the induction of an inflammatory response via TLR-2 or TLR-4 (Ttoll-like receptor-4), thus playing a critical role in acne inflammation (36). Davidsson et al. found that *Cutibacterium acnes* contributes to an immunosuppressive environment in prostate cancer by recruiting Tregs and increasing the expression of immunosuppressive mediators such as PD-L1 (Programmed cell death 1 ligand 1), CCL17 (C-C motif chemokine ligand 17), and CCL18 (C-C motif chemokine ligand 18) (37). High immunosuppression is a vital characteristic of PDAC, and this result suggests that *Cutibacterium acnes* may contribute to the formation of this immunosuppressive microenvironment of PDAC. The abundance of gut commensal bacteria *Dialister hominis*, a succinate consumer, was reduced in PDAC tumor tissues compared to non-tumor tissues, which differs from previous findings in digestive tract cancer (38, 39). A meta-analysis based on six studies showed that the relative abundance of *Dialister* was significantly higher in mucosal from gastric cancer patients than in control samples (38). In addition, a high abundance of the genus *Dialister* in tumors, adjacent tumors, and off-tumor areas was associated with shorter overall survival in colorectal cancer patients and works as an index for predicting the risk of colorectal cancer recurrence and disease prognosis (39). This discrepancy in abundance may be attributed to variances in the host organ being colonized or may stem from the finer taxonomic resolution employed in this study, which extends to the species level.

In our survival analysis, we discovered that patients whose tumor was colonized with *Ralstonia pickettii_B* had significantly worse OS. *Ralstonia pickettii* is a non-fermenting gram-negative commensal bacillus and is an opportunistic pathogen that often causes nosocomial infections (40). However, the effects of *Ralstonia pickettii* in tumorigenesis have yet to be well studied. A study by Higuchi et al. observed that *Ralstonia pickettii* was presented in almost all mesothelioma patients (41). In addition, *Ralstonia pickettii*

belongs to class Gammaproteobacteria, which has been reported to carry long cytidine deaminase that can metabolize the chemotherapy drug gemcitabine, thereby inducing chemoresistance in pancreatic cancer (12). Therefore, induction of chemoresistance may be the mechanism by which colonization of *Ralstonia pickettii* significantly shortens the overall survival time of pancreatic cancer patients after surgery.

After conducting two untargeted approaches, our current metabolomic analysis revealed a significantly higher number of decreasing lipid and lipid-like compounds than increasing ones in the PDAC group. Most of these compounds could be classified into the fatty acyls and glycerophospholipids classes. Dysregulated lipid metabolism is now recognized as a hallmark of many malignancies (42, 43). High phosphorylcholine and low glycerophosphorylcholine levels are consistently observed in aggressive cancers, and an elevated phosphorylcholine/glycerophosphorylcholine ratio has also been proposed as a biomarker of tumor progression (44–46). MPEA further implied that glycerophospholipid metabolism is a critical pathway in PDAC pathogenesis. In the present study, substrate sn-glycero-3-phosphoethanolamine and its product G3P were downregulated in PDAC samples, while metabolic enzymes glpQ and ugpQ originating from bacteria were significantly enriched in PDAC. This phenomenon may be attributed to the high demand for G3P consumption in pancreatic cancer metabolism since the glycerol-3-phosphate shuttle serves as a crucial NADH shuttle mechanism, not only facilitating the transfer of cytosolic reducing equivalents into the mitochondria but also acting as a metabolic hub linking glycolysis, lipid synthesis, and oxidative phosphorylation (47).

By integrating multi-omics data, our study revealed a range of microbiome-metabolite interactions. Association analysis indicated a markedly reduced number of statistically significant correlations between microbial species and metabolites within PDAC tumor samples, as opposed to the NAT samples, which coincides with the decrease in bacterial α-diversity in PDAC progression. Within the module of amino acid metabolism, MPEA identified key metabolic routes, including alanine, aspartate and glutamate metabolism, and arginine biosynthesis. Notably, there was an observed enrichment of beta-alanine, coupled with a marked depletion of L-aspartic acid, L-glutamine, and L-serine in PDAC. These alterations could be attributed to pancreatic microbiota variations and their associated enzymatic activities. Furthermore, the enzyme K00823 [EC:2.6.1.19] and its substrates beta-alanine and 2-oxoglutarate demonstrated significant enrichment in PDAC. Beta-alanine was reported to suppress tumor aggressiveness *in vitro* (48). Simultaneously, the PDAC-enriched *Staphylococcus aureus* exhibited a significant positive correlation with K00823 while showing a marked negative correlation with the anti-tumor potential metabolite beta-alanine, suggesting that *Staphylococcus aureus* may influence the tumorigenesis of PDAC through its involvement in amino acid metabolism. However, due to the potential of beta-alanine as a dietary supplement, the absence of dietary information collected from patients in this study does not preclude the possibility of beta-alanine originating from diet. Moreover, Vaughan et al. have highlighted the significant role of β-alanine in regulating cytoplasmic acidity (48). Another notable finding in this study is the representative reaction involving the decomposition of carbonate catalyzed by bacterial enzyme EC:2.7.7.87, suggesting that this reaction may also contribute to the regulation of acidity within the tumor microenvironment.

Our study has several limitations. Although 2bRAD-M is very powerful in characterizing microbiota at the species level and covering a comprehensive range of species, it has a limitation in uncovering the full genetic content compared to metagenomic sequencing. We found that the M stage significantly impacted certain microbes, but only three M1 stage patients in our study cohort required future exploration of advanced PDAC. Additionally, our study used pancreatic tissues for metabolomic detection. Due to the complexity of the PDAC tumor microenvironment, the detection results not only characterized the metabolic alteration of PDAC tumor cells but also provided a picture of the entire microenvironment.

In conclusion, leveraging multi-omics data, our study attempted to reveal the ordinary states of pancreatic microbiome dysbiosis and metabolome dysregulation in patients with PDAC. We found that microbial species affect the tumorigenesis, metastasis, and prognosis of PDAC and identified unique microbe-enzyme-metabolite interaction. Although more mechanistic studies and clinical validation are needed, our study can provide a novel insight for the need to investigate the potential associations between pancreatic microbiota-derived omics signatures, which may drive the clinical transformation of microbiome-derived strategies toward therapy-targeted bacteria.

## MATERIALS AND METHODS

### Study participants and sample collection

A total of 105 patients diagnosed with PDAC who underwent surgery between July 2016 and August 2022 at the Peking Union Medical College Hospital, Peking, China, were enrolled for microbiome and untargeted metabolome analysis. Normal adjacent tissues were used as controls. All diagnoses were made by postoperative pathological examinations. Tissue samples were collected during surgery into a sterile tube which were then stored at −80°C for microbiome sequencing and metabolic analysis. The tumor stage was evaluated based on the TNM staging system. Tumor differentiation was assessed using the standard pathological grading scheme into well-differentiated, moderately differentiated, or poorly differentiated based on the lowest differentiation grade observed.

### Microbiota analysis

A detailed description of DNA extraction, amplification, sequencing processing, and decontamination has been provided in the supplementary materials. In brief, the genomic DNA of pancreatic tissues was extracted using a TIANamp Micro DNA Kit (Tiangen, cat. #DP316). The 2bRAD-M library preparation method was primarily based on the original protocol developed by Wang et al. (49, 50), with slight modifications. The PCR products were purified using a QIAquick PCR purification kit (Qiagen) and then sequenced using the Illumina Nova PE150 platform. 2bRAD-M sequencing was performed at Qingdao OE Biotech Co., Ltd. (Qingdao, China). Reads with an N base proportion greater than 8% and low-quality reads (with a base quality value below Q30 constituting more than 20% of the total bases in a read) were removed during sequence quality control. To identify the microbial species within each sample, the sequenced 2bRAD tags underwent quality control and were subsequently mapped against the 2bRAD marker database using a built-in Perl script (50). The relative abundance of a specific species was determined by calculating the ratio of the number of microbial individuals attributed to that species to the total number of individuals from known species detectable within a given sample. The present study employed air samples from laboratory and surgical environments as negative controls, utilizing a combination of decontam, microDecon, and FEAST to eliminate background microorganisms from the experimental samples effectively. Contaminants were identified based on tissue samples and environmental control samples using microDecon's decon function and decontam's isContaminant function. Based on the list of contaminants identified by microDecon and decontam, the union set is taken as the final pollutant list. The proportion of unknown origin calculated using FEAST replaces the value of the contaminant in the experimental group sample. Each experimental sample was normalized by dividing by the sum of the samples to obtain the relative abundance after decontamination. Decontamination analyses used default parameters. The relative abundance feature table was imported into R for further analysis. The α-diversity of each sample was evaluated using the Chao 1, Simpson, and Shannon indices calculated on the species level. Compositional differences between each pair of groups were analyzed using PERMANOVA (999 permutations). The distance matrix was constructed based on the Bray-Curtis distance of the relative

abundance of species. The compositional shift was visualized using PCoA based on the same distance matrix. The alterations at genus, species, and KO gene levels among different groups were determined by MaAsLin2 (Microbiome Multivariable Associations with Linear Models, MaAsLin2 R package) with BMI adjusted according to clinical details of included patients. Species with a total relative abundance greater than 0.05% and a prevalence of greater than 10% were included in differential analysis. The significance criteria were prevalence >10% and adjusted $Q$ value <0.25 as default (26).

## Survival analysis

Survival analysis was conducted using PDAC samples. Overall survival includes death from any cause as events after the perioperative period. Person-time refers to the duration from surgery to the occurrence of event or loss to follow-up (censored) for all endpoints. Microbial species with relative abundance under 0.05% and prevalence under 10% were excluded, resulting in the inclusion of 26 species. Clinical characteristics [age, gender, BMI level, smoking, alcohol, family history, other malignancy, diabetes, hyperlipidemia, antibiotics usage, location, upregulation of CA19-9, differentiation, perineural invasion, blood vessel invasion, T stage, N stage, M stage, stage, and neoadjuvant therapy] were included in the model. We built an elastic-net penalized Cox regression model using the "glmnet" function in the "glmnet" R package, with an α value of 0.5 to allow groups of correlated predictors to be selected. A 100 times 10-fold cross-validation for the elastic-net penalized Cox regression was conducted using the "cv.glmnet" function to determine the value of optimal lambda.1se to build a regularized Cox model with the fewest number of variables. We summed the number of times each factor was selected out of the 100 repetitions. We focused further on factors selected ≥50% of the 100 times (50 times or more) and with $P < 0.20$ in standard univariate Cox proportional hazards models. The effects of identified species and clinical characteristics on OS were investigated using Kaplan-Meier survival curves, and compared using the log-rank test.

## Metabolome data analysis

A detailed description of sample preparation, experiment condition, and data processing for LC-MS and GC-MS untargeted metabolome analysis has been described in the supplementary materials. The quality control results are shown in Fig. S7. In brief, the original LC-MS data were processed using Progenesis QI V2.3 (Nonlinear, Dynamics, Newcastle, UK) for baseline filtering, peak identification, integration, retention time correction, peak alignment, and normalization. The GC/MS rawdata were obtained in .D format and were transferred to .abf format using the software Analysis Base File Converter. The data were then imported into MS-DIAL software, which performs peak detection, peak identification, MS2Dec deconvolution, characterization, peak alignment, wave filtering, and missing value interpolation. After the data were normalized, redundancy removal and peak merging were performed to obtain the data matrix.

The matrix was imported into R for analysis. OPLS-DA was used to identify differentiating metabolites between the groups. To mitigate overfitting, sevenfold cross-validation and 200 response permutation testing were performed to evaluate the model's quality. VIP value derived from the OPLS-DA model was used to rank the overall contribution of each variable to group discrimination. Subsequently, a two-tailed Student's $t$-test was conducted to verify the statistical significance of the identified metabolites differentiating between the groups. Differential metabolites were selected based on VIP values greater than 1.0 and $P$-values less than 0.05.

## Analysis of microbiome-metabolites interactions

Metabolite origin was analyzed using MetOrigin (51). Spearman correlation analysis was performed using the "psych" package in R to investigate the associations between differential microbial species, microbial KO genes, and microbe-derived metabolites in

PDAC and NAT, respectively. Only associations with an absolute correlation coefficient ($R$) value greater than 0.3 and $P < 0.05$ were considered significant. The resulting associations were visualized using heatmap and network.

## Statistical analyses

All pairwise comparisons were performed using a two-sided Wilcoxon rank-sum test (Mann-Whitney $U$-test). Dissimilarity tests among groups (PERMANOVA) were conducted on Euclidean distance for metabolites and Bray-Curtis distance for bacteria, with 999 permutations in the R package vegan. Multiple comparisons were adjusted using Benjamini-Hochberg method. All statistical analyses were performed using R, version 4.2.1.

## ACKNOWLEDGMENTS

The authors would like to thank Professor Taiping Zhang for his efforts in sample collection. Wenbo Kou of OE Biotechnology Company (Qingdao, China) for his technical support in removing contamination from the sequencing data. This study was funded by National Multidisciplinary Cooperative Diagnosis and Treatment Capacity Building Project for Major Diseases, National High Level Hospital Clinical Research Funding (2022-PUMCH-D-001), Chinese Academy of Medical Sciences (CAMS) Innovation Fund for Medical Sciences (CIFMS, 2021-I2M-1–002), National Multidisciplinary Cooperative Diagnosis and Treatment Capacity Building Project for Major Diseases (NSFC, 81970763), Nonprofit Central Research Institute Fund of CAMS (PT201832014), National Natural Science Foundation of China (NSFC, 82103016) and (NSFC, 62133006). The funders played no role in study design, data collection, analysis and interpretation of data, or the writing of this manuscript.

L.Y. and Y.P.Z. supervised the study. J.R., X.W., Y.F. and F.H.Z. collected samples and provided clinical information. Y.F. performed the bioinformatics analyses. S.H. have provided helpful comments. Y.F. and X.H.L. interpreted the results. Y.F. wrote the manuscript.

All authors have read and approved the final manuscript.

## AUTHOR AFFILIATIONS

[1]Department of General Surgery, Peking Union Medical College Hospital, Peking Union Medical College, Chinese Academy of Medical Sciences, Beijing, China
[2]Key Laboratory of Research in Pancreatic Tumor, Chinese Academy of Medical Sciences, Beijing, China
[3]National Science and Technology Key Infrastructure on Translational Medicine in Peking Union Medical College Hospital, Beijing, China
[4]State Key Laboratory of Complex Severe and Rare Diseases, Peking Union Medical College Hospital, Chinese Academy of Medical Sciences and Peking Union Medical College, Beijing, China
[5]Faculty of Dentistry, The University of Hong Kong, Hong Kong SAR, China

## AUTHOR ORCIDs

Shi Huang http://orcid.org/0000-0002-7529-2269
Lei You http://orcid.org/0000-0001-6030-6560
Yupei Zhao http://orcid.org/0000-0001-7081-2299

## FUNDING

| Funder | Grant(s) | Author(s) |
|---|---|---|
| National High Level Hospital Clinical Research Funding | 2022-PUMCH-D-001 | Lei You |

| Funder | Grant(s) | Author(s) |
|---|---|---|
| CAMS \| Chinese Academy of Medical Sciences Initiative for Innovative Medicine (中国医学科学院创新工程) | 2021-I2M-1-002 | Lei You |
| National Multidisciplinary Cooperative Diagnosis and Treatment Capacity Building Project for Diseases | 81970763 | Lei You |
| Nonprofit Central Research Institute Fund of CAMS | PT201832014 | Lei You |
| MOST \| National Natural Science Foundation of China (NSFC) | 82103016 | Lei You |
| MOST \| National Natural Science Foundation of China (NSFC) | 62133006 | Lei You |
| National Multidisciplinary Cooperative Diagnosis and Treatment Capacity Building Project for Major Diseases | Not applicable | Lei You |

## AUTHOR CONTRIBUTIONS

Yuan Fang, Conceptualization, Formal analysis, Investigation, Visualization, Writing – original draft | Xiaohong Liu, Visualization, Writing – review and editing | Jie Ren, data curation, Resources, Writing – review and editing | Xing Wang, data curation, Resources, Writing – review and editing | Feihan Zhou, data curation, Resources, Writing – review and editing | Shi Huang, Methodology, Writing – review and editing | Yupei Zhao, Project administration, Resources, Supervision.

## DATA AVAILABILITY

The data sets required to reproduce the results in the current study are included in this published article and its supplementary information files. No unique code was generated in this study. The code and raw sequencing data for this study is available upon request from corresponding author Lei You, florayo@163.com.

## ETHICS APPROVAL

Ethical approval for this study was obtained from the Human Research Ethics Committee of Peking Union Medical College Hospital I-23PJ1417. The requirement for consent was waived by the ethics committee due to the retrospective study.

## ADDITIONAL FILES

The following material is available online.

### Supplemental Material

**Dataset S1 (Spectrum00962-24-S0001.xlsx).** Dataset required to reproduce the results in the manuscript.
**Figure S1 (Spectrum00962-24-S0002.pdf).** Stackplot of bacterial composition.
**Figure S2 (Spectrum00962-24-S0003.pdf).** Microbial biomarkers identified by LEfSe analysis.
**Figure S3 (Spectrum00962-24-S0004.pdf).** Differential bacterial species associated to tumor metastasis.
**Figure S4 (Spectrum00962-24-S0005.pdf).** Subgroup survival Kaplan-Meier curve Presence of *Ralstonia pickettii_B*.
**Figure S5 (Spectrum00962-24-S0006.pdf).** Heatmap of differential metabolites.
**Figure S6 (Spectrum00962-24-S0007.pdf).** Correlation between differential metabolites.
**Figure S7 (Spectrum00962-24-S0008.pdf).** PCA plot of metabolome quality control.
**Supplemental material (Spectrum00962-24-S0009.docx).** Legends for supplemental figures and tables, and additional experimental details.
**Table S1 (Spectrum00962-24-S0010.docx).** The STORMS checklist.

## Open Peer Review

**PEER REVIEW HISTORY (review-history.pdf).** An accounting of the reviewer comments and feedback.

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
