## [Reviewer comments · Microbiology Spectrum]

Microbiology Spectrum

Integrated Analysis of Microbiome and Metabolome Reveals Signatures in PDAC Tumorigenesis and Prognosis

Lei You, Yuan Fang, Xiaohong Liu, Jie Ren, Xing Wang, Feihan Zhou, Shi Huang, and Yu-Pei Zhao

Corresponding Author(s): Lei You, Peking Union Medical College Hospital Department of General Surgery

Review Timeline:

Submission Date:	April 29, 2024
Editorial Decision:	May 19, 2024
Revision Received:	May 28, 2024
Editorial Decision:	August 12, 2024
Revision Received:	August 22, 2024
Accepted:	September 16, 2024

Editor: Pei-Yuan Qian

Reviewer(s): Disclosure of reviewer identity is with reference to reviewer comments included in decision letter(s). The following individuals involved in review of your submission have agreed to reveal their identity: Yixuan Meng (Reviewer #1); Xun Lan (Reviewer #4)

Transaction Report:

DOI: <https://doi.org/10.1128/spectrum.00962-24>

Re: Spectrum00962-24 (Integrated Analysis of Microbiome and Metabolome Reveals Signatures in PDAC Tumorigenesis and Prognosis)

Dear Dr. Lei You:

Thank you for the privilege of reviewing your work. Below you will find my comments, instructions from the Spectrum editorial office, and the reviewer comments.

Revision Guidelines

Sincerely,
Pei-Yuan Qian
Editor
Microbiology Spectrum

Reviewer #1 (Comments for the Author):

The manuscript submitted by the authors presents an exploration of the association between the microbiome, metabolic substances, and the development of pancreatic ductal adenocarcinoma (PDAC).

However, there are several areas in the manuscript that require attention to ensure the clarity and robustness of your study

findings.

Figure 3A - Presentation of Incidental Results:

It is unusual to include incidental findings in the main figures of a manuscript as this might distract from the main messages of the study. Consider moving these results to supplementary materials unless they are directly relevant to the main research questions addressed in your paper.

Methods Section - Survival Analysis:

Please specify how many times the 10-fold cross-validation was repeated in your survival analysis. The selection of *Ralstonia pickettii_B* needs clearer justification. Out of the 26 microbial species and clinical characteristics evaluated, Detail the selection criteria and the statistical thresholds used for including this species in your model.

Page 10, Line 194 - Kaplan-Meier Curve and Survival Analysis:

You mention the use of Kaplan-Meier curves tested by the log-rank test. While this is a valid approach for univariate survival analysis, the relationship and comparison with Cox proportional hazards regression results should also be clarified. Please include the results of the standard Cox regression and the outputs from the repeated 10-fold cross-validation for the 26 species.

Please thoroughly revise the manuscript to correct all grammatical errors and enhance the clarity of arguments. Including "For α -diversity, significant decrease was observed in the in PDAC" in line 141; "On the one hand, if this is a limit, the comparison of matched tumor and NAT areas minimized..." on page 15, line 296. et al.

Reviewer #2 (Comments for the Author):

In this manuscript, Fang et al. investigated the pancreas tissue microbiome and metabolome in pancreatic ductal adenocarcinoma (PDAC), using 103 PDAC tumor samples and 105 adjacent normal samples collected from PDAC patients. They identified differences in overall microbial diversity and composition, as well as individual microbial taxa, and microbiome-derived metabolites between tumor and adjacent normal tissues, some of which are associated with survival.

The main strength of this study is the large sample size of tissue samples from patients of PDAC, a lethal disease, and the implementation of comprehensive metabolome profiling by the combination of untargeted liquid chromatography-tandem mass spectrometry (LC-MS), and untargeted gas chromatography-mass spectrometry (GC-MS). Further, metabolite origin analysis made it able to explore the potential microbiome-host interactions in PDAC development.

My major concerns are listed below.

- (1) The manuscript was badly written. Many abbreviations are used before spelling out. The digits of numbers are not consistent. Many microbial taxa names are not italicized. Unless comprehensive English editing was performed by a local writer, the writing quality of this work does not align with the average levels of paper published in mSystems.
- (2) I'm quite concerned about the soundness of the microbiome profiling technology used in this study. The original paper describing this technology was published more than 20 years ago and has been rarely used in microbiome studies in the past 5 years. Also, the authors described that they identified a total of 1,920 species but only 32 of them retained for differential abundance analyses after filtering low prevalence and abundance taxa. If this is not a typo, it means that >98% of species are of low prevalence or abundance. I do not think a technology is sound if >98% signals it identifies are of low quality. Unless this concern can be fully addressed, I can not endorse the publication of this work.
- (3) The analysis of genus-level taxa does not make sense to me because it was derived using only 32 species. This derived genus-level information deviated a lot from the real genus-level information.
- (4) "The 179 model included group and M stage as fixed effects and BMI as a random effect." - I do not think this is the correct usage of MaAsLin 2. Fixed and random effect variables are incorrectly assigned.
- (5) It is not clear how PICRUSt2 was used to infer functional features. If the authors use the 32 species, it is hard to believe 6,083 KO genes could be identified. While, if the 1,920 species was used, it does not make sense because >98% of these species are of low prevalence/abundance.
- (6) Negative controls were used in DNA extraction and PCR amplification process. However, it is not clear whether microbial profiling was performed for negative control samples. *Cutibacterium acnes* stands out as one of the differentially abundance species. This species is more commonly seen in the skin microbiota and linked to acne. This species might be derived from contamination.
- (7) It is not clear whether the patients underwent treatment and the type of treatment. Treatment should be an important covariant that should be adjusted in all analyses.

**Integrated Analysis of Microbiome and Metabolome Reveals**

**Signatures in PDAC Tumorigenesis and Prognosis**

Yuan Fang^{1234#}, Gang Yang^{1234#}, Bo Ren^{1234#}, Xiaohong Liu¹²³⁴, Jie
Ren¹²³⁴, Xing Wang¹²³⁴, Feihan Zhou¹²³⁴, Minzhi Gu¹²³⁴, Ruiling Xiao¹²³⁴,
Jialu Bai¹²³⁴, Xiyuan Luo¹²³⁴, Shi Huang⁵, Lei You^{1234*}, Taiping Zhang
6 ^{1234*} and Yupei Zhao^{1234*}

**Author affiliations and addresses:**

1.Department of General Surgery, Peking Union Medical College Hospital, Peking
Union Medical College, Chinese Academy of Medical Sciences, 100730, Beijing,
China.

2.Key Laboratory of Research in Pancreatic Tumor, Chinese Academy of Medical
Sciences, 100730, Beijing, China.

3.National Science and Technology Key Infrastructure on Translational Medicine in
Peking Union Medical College Hospital, 100730, Beijing, China.

4.State Key Laboratory of Complex Severe and Rare Diseases, Peking Union Medical
College Hospital, Chinese Academy of Medical Sciences and Peking Union Medical
College, 100730, Beijing, China.

5.Faculty of Dentistry, The University of Hong Kong, 999077, Hong Kong SAR,
China

**E-mail addresses:**

- Yuan Fang, 985311609@qq.com
- Gang Yang, doc.gang@qq.com
- Bo Ren, berserker94@163.com
- Xiaohong Liu, Liu_xh1001@163.com
- Jie Ren, 18813126010@163.com
- Xing Wang, drwang99@163.com
- Feihan Zhou, 15261588335@163.com
- Minzhi Gu, gmz1433@163.com
- Ruiling Xiao, xiaorl66@163.com
- Jialu Bai, 3277005650@qq.com
- Xiyuan Luo, b2023001097@pumc.edu.cn
- Shi Huang, shihuang@hku.hk
- ***Authors for correspondence:**
- Lei You, florayo@163.com; Tel: +86-10-69156007. No. 1 Shuaifuyuan, Wangfujing
- Street, Beijing 100730, China

- Taiping Zhang, tpingzhang@yahoo.com; Tel: +86-10-69156007. No. 1 Shuaifuyuan,
Wangfujing Street, Beijing 100730, China
- Yupei Zhao, E-mail: zhao8028@263.net; Tel: +86-10-69156007; Fax:
+86-10-65124875; No. 1 Shuaifuyuan, Wangfujing Street, Beijing 100730, China

**Abstract**

Emerging evidence suggests the involvement of the microbiome and metabolic substances in
the development of pancreatic ductal adenocarcinoma (PDAC), yet the results remain
contradictory and there is limited research on the correlation between microbiome and
metabolites in PDAC. We aimed to identify the functional differential profile of the
pancreatic microbiome and metabolites in patients with PDAC. We collected pancreatic tissue
samples from 105 PDAC patients and performed a 6-year follow-up. One hundred and five
normal adjacent tissue (NAT) samples and 103 tumour samples were collected. 2bRAD-M
sequencing, untargeted liquid chromatography-tandem mass spectrometry (LC-MS), and
untargeted gas chromatography-mass spectrometry (GC-MS) were performed on these paired
samples. Compared with NATs, microbial alpha diversity decreased in tumour tissues. The
PDAC tumour had higher relative abundance of *Staphylococcus aureus*, *Cutibacterium acnes*
and *Cutibacterium granulosum* after adjusting for confounding factors BMI and M stage, and
the presence of *Ralstonia pickettii_B* was associated to a worse overall survival. Metabolomic
analysis revealed distinctive differences in composition between PDAC and NATs, with 417
and 147 discriminative metabolites identified using LC-MC and GC-MS, respectively.
Metabolites origin analysis identified 145 differential metabolites related to microbiota. Most
of these were lipids and lipid-like molecules. Importantly, alterations of these metabolites
were found to be correlated with changes in the relative abundance of PDAC-related bacterial
species via KO genes. Microbial dysbiosis in PDAC is associated with decreased diversity
and alterations in bacteria involved in an altered microbiota-metabolism interplay.

**Importance**

We conducted a large sample size pancreatic adenocarcinoma microbiome study
using a novel microbiome sequencing method and two metabolomic assays. Two
significant outcomes of our analysis are (a) Commensal opportunistic pathogens
*Staphylococcus aureus*, *Cutibacterium acnes* and *Cutibacterium granulosum* were
enriched in PDAC tumours compared with normal adjacent tissues, and worse overall
survival was found related to presence of *Ralstonia pickettii*_B. (b). Microbial species
affect the tumorigenesis, metastasis, and prognosis of PDAC via unique microbe–
enzyme-metabolite interaction. Thus, our study highlights the need for further
investigation of the potential associations between pancreatic microbiota-derived
omics signatures, which may drive the clinical transformation of microbiome-derived
strategies towards therapy-targeted bacteria.

**Keywords:** PDAC; pancreatic ductal adenocarcinoma; microbial metabolism; microbiota;
metabolome; carcinogenesis

**Introduction**

Pancreatic ductal adenocarcinoma (PDAC) remains a highly fatal malignancy,
with a 5-year overall survival of 12%⁽¹⁻³⁾. Therapeutic methods for PDAC remain
limited⁽⁴⁾. Surgery combined with adjuvant chemotherapy remains the only curative
therapeutic option, but more than 60% of patients are diagnosed with unresectable

disease⁽⁵⁾. The pathogenesis of PDAC is complex, and genetic alterations in PDAC
fail to explain carcinogenesis alone, which leaves environmental factors including the
microbiota emerging as potential mediators of PDAC carcinogenesis. Thus, as an
important hallmark of cancer⁽⁶⁾, the role of polymorphic microbiomes in cancer
remains to be explored, and there is a pressing need to identify microorganisms that
might explain the differences between PDAC tumours and normal pancreatic tissue so
that new concepts can be developed for future therapies⁽⁷⁾.

Numerous studies have shown that microorganisms play a critical role in the
carcinogenesis^(8, 9). Intratumor bacteria have been observed in various tumours,
including PDAC⁽¹⁰⁾, which is associated with PDAC carcinogenesis, progression, and
poor prognosis via a complex mechanism. Geller et al. initially proposed a connection
between intratumoural microbiota and PDAC⁽¹¹⁾. The prevalence of intratumoural
bacteria in PDAC tissues was significantly higher than that in normal pancreatic
tissues. The presence of microbiota in the pancreas of both healthy and cancerous
subjects was also confirmed in a study by Thomas et al.⁽¹²⁾. In a study involving 12
PDAC patients, Pushalka et al. found higher bacterial biomass in PDAC tumours than
in normal pancreatic tissue^(13, 14). Alpha diversity was reported to be slightly higher in
healthy controls versus in patients⁽¹⁵⁾. The most common class identified in the PDAC
intratumor microbiome is Gammaproteobacteria, with the dominant genus
*Pseudomonas*^(11, 13), which carries long-form cytidine deaminase (CDDL) that
metabolizes the chemotherapeutic drug gemcitabine (2',2'-difluorodeoxycytidine) into
its inactive form (2',2'-difluorodeoxyuridine)⁽¹¹⁾. Riquelme et al. investigated the

impact of tumour microbiota on PDAC patient survival⁽¹⁶⁾. Patients with long-term
survival exhibited higher alpha diversity and enrichment for *Pseudoxanthomonas*,
*Saccharopolyspora*, and *Streptomyces*. Guo et al. revealed that *Acinetobacter*,
*Pseudomonas*, and *Sphingopyxis*, as the intratumoural microbiota of basal-like
PDAC, were associated with worse prognosis by inducing inflammation⁽¹⁷⁾. Taken
together, these data support that distinctive profiles of tumour microbiota may
underlie PDAC heterogeneity, and comprehensive characterization of the PDAC
intratumor microbiome may be an essential step in unraveling the effects of bacteria
on PDAC tumourigenesis and prognosis. Despite these developments, the clinical
significance of the intratumoural microbiome in PDAC is still poorly understood.
Previous comparative PDAC-healthy control studies generally constrained by
limitations of small number of samples and vague taxonomy classification, thus, large
sample size studies are urgently required. Therefore, it is imperative to further
investigate the profiles of the PDAC tumour microbiome and clarify its clinical
significance and prognostic value.

Microbiota-derived metabolites are important natural products that establish a
strong connection between the microbiome and cancer⁽¹⁸⁾. For example, microbial
byproducts can actively contribute to carcinogenesis. Secondary metabolites,
including lithocholic acid and deoxycholic acid^(19, 20), as well as catabolites, such as
acetate and butyrate^(21, 22), play a crucial role in enhancing either epithelial-
mesenchymal transition or cell proliferation in several models of cancer⁽²³⁾.
Metabolomic comparisons of human PDAC tumour tissue and normal adjacent tissue

(NAT) revealed that tumour tissues exhibit lower levels of glucose, upper glycolytic
intermediates, creatine phosphate, and the amino acids glutamine and serine, which
are the major metabolic substrates⁽²⁴⁾. However, evidence of the involvement of
microbiome-derived metabolites in PDAC carcinogenesis is limited.

In summary, to advance our understanding of the microbiome and
metabolome characteristics associated with PDAC and to elucidate the intricate role
of their interaction in PDAC carcinogenesis and prognosis, we conducted
comprehensive analyses of the microbiome and metabolome of surgically excised
PDAC tumor and its matched NATs from a large scale of 105 patients based on
2bRAD-M sequencing, untargeted liquid chromatography-tandem mass spectrometry
(LC-MS), and untargeted gas chromatography-mass spectrometry (GC-MS). Our
study provided data support for subsequent studies on PDAC, thereby expanding the
perspective in this field.

**Results**

*Participant characteristics*

We collected 208 tissue samples (103 paired PDAC and NATs, plus unpaired 2
NATs) from 105 patients. The demographic, clinical, and histopathological
characteristics of the patients are shown in Table 1. The average age was 61.57 ± 8.18 ,
and 63 patients (60.0%) were male. Diabetes was present in 39 (37.1%) patients, and
among them 14 (13.3%) were new-onset diabetes. The number of patients with TNM

stages I, II, III and IV were 36 (34.3%), 47 (32.4%), 19 (18.1%) and 3 (2.9%),
respectively. Tumour differentiation was well (16.1%), moderate (41.0%) and poor
(42.9%).

***PDAC and NAT microbiome differ from global scale***

Negative controls were used in the DNA extraction and PCR amplification
processes to remove the interference of contaminating microorganisms introduced
during the experimental manipulation. Using a combination of decontam,
microDecon, and FEAST, background microorganisms in the samples were deducted
based on the negative controls. After decontamination, 1920 species were identified.
A broad overview of our taxonomic data from the 105 subjects is provided in
Supplementary Figure 1.

To determine the differences in pancreatic tissue microbiota associated with
primary PDAC tumours, we analysed the α -diversity metrics between PDAC and
NAT. Significant decrease was observed in the Chao1, Shannon, and Simpson indices
in PDAC (Figure 1A; $p = 0.0017, 0.00012, \text{ and } 0.00089$, respectively). Given the
significant differences in α -diversity between NAT and PDAC tissues, we explored
the compositional diversity of the microbiota in these two groups. A Venn diagram of
the microbiota composition showed that NAT had a greater variety of microorganisms
(Figure 1B). At the species level, 512 microorganisms were shared by the NAT and
PDAC groups. The Bray-Curtis distance was used to perform principal coordinate
analysis (PCoA) of the samples (Figure 1C). The results of the PERMANOVA test

indicated a statistically significant difference in the β -diversity between PDAC and
NAT ($R^2 = 0.011$; $p = 0.01$).

Next, we sought to identify the potential factors associated with microbial
profiles. Adonis was used to explore the associations between variations in the
pancreatic microbiota and host characteristics. Given a false discovery rate (FDR) of
5%, three parameters were significantly associated with microbial variations derived
from Bray–Curtis distances calculated on species level (Figure 1C). Group, M stage,
and BMI level were explanatory factors, which is consistent with previous research.

***Taxonomic signatures of microbiota in PDAC tumor***

Next, we attempted to identify PDAC-associated taxa. After filtering out low
prevalence (<10%) and abundance (<0.1%) taxa, a total of 32 species remained for
differential abundance analysis. We then determined the signatures associated with
PDAC in the most abundant taxonomic groups using multivariable microbiome
associations with a linear model (MaAsLin 2) to control for confounding factors. The
model included group and M stage as fixed effects and BMI as a random effect.
Thirteen species between the PDAC and NAT groups were identified as differentially
abundant bacterial species (Figure 2A and 2C). Samples from the PDAC group were
enriched in *Staphylococcus aureus*, *Cutibacterium acnes* and *Cutibacterium*
*granulosum*, whereas depleted in *Sphingomonas aquatilis*, *BACL27 sp014190055*,
*QWOQ01 sp003669585*, *Limnohabitans_A sp005789685*, *Mycobacterium koreense*,
*Mycobacterium intermedium*, *UBA953 sp002293125*, *Bacillus_A bombysepticus*,

*Pelomonas* sp003963075 and *Dialister hominis*. After adjusting for M stage, the
associations were still significant. To compare our results with previous studies, we
performed difference analysis at the genus level, the results were broadly consistent
with the species level. Besides, *Bifidobacterium* was slightly increased in PDAC,
while *Dietzia* and *Streptococcus* were depleted (Figure 2B and 2D).

The pathogens *Pseudomonas fulva*, *Dietzia maris*, *Massilia timonae* and
*Brevundimonas diminuta* were positively associated with the M1 stage, whereas
*Pseudomonas_E* sp900187635 was depleted in the M1 stage (Figure 3A). In a subset
containing only PDAC samples, these results still hold. However, it is important to
note that only three patients were diagnosed with stage M1, rendering this result
incidental.

To investigate the function of the intratumoural microbiota in PDAC, we
predicted the biological functions of the bacteria utilizing PICRUSt2. We identified
6083 KO genes altogether. Using the MaAsLin 2 analysis, a total of 1073 KO genes
were found to be differentially expressed between the two groups after adjusting for
BMI.

***Microbial species related to overall survival***

Analysis of the relationship between microbial species and overall survival is
performed only in PDAC tumour samples. Among them, one patient with
perioperative cardiac death and four patients with uncertain time of death were
excluded, a total of 98 PDAC patients were included in the survival analysis, resulting

in a median of 15 months of follow-up (range 1–71 months).

In 10-fold cross-validated elastic-net Cox regression models for OS, we found
species *Ralstonia pickettii_B* and age were associated with OS. Based on these results,
we then tested the relationship between *Ralstonia pickettii_B* and OS by stratifying
the patients in two groups based on occurrence of *Ralstonia pickettii_B*. As expected,
we found that patients colonized with *Ralstonia pickettii_B* had significantly worse
OS (median OS: 17 months) than those *Ralstonia pickettii_B* negative ones (median
OS: 37 months) using Kaplan-Meier curve tested by log-rank ($p = 0.045$) (Figure 3D).
Given that PDAC is a disease in which risk increases with age, we stratified the
patients into two groups by age 65. A median OS of 36 months was obtained for
middle-aged group and 17 months for elder patients group ($p = 0.047$) (Figure 3E).
Our findings indicate that the relative abundance of *Ralstonia pickettii_B* in tumour
could serve as a predictor of survival outcome in resected PDAC patients, further
elaborated the potential of the microbiome composition in mediating PDAC
progression.

***Untargeted metabolomics profiling revealed significantly altered metabolites***

To investigate the interactions between the pancreatic microbiota and
host-microbe co-metabolism, untargeted metabolomic profiling was performed on a
subset of 98 NAT and 90 PDAC samples. We conducted both LC-MS and GC-MS to
make our assay as comprehensive as possible. A total of 6375 metabolites were
quantified from tissue samples using LC-MS and 481 were quantified using GC-MS.

Orthogonal partial least squares-discriminant analysis (OPLS-DA) (Figure 4A
and 4B) showed that there were differences in the tissue metabolite profiles between
PDAC and NAT groups, indicating a tumor-metabolite shift in PDAC carcinogenesis.
The ability of the OPLS-DA model was tested during a seven-cross validation through
200 random permutation tests. The intercepts of goodness-of-fit (R²) and
goodness-of-prediction (Q²) illustrate that the OPLS-DA model is reliable and does
not overfit. We plotted fold changes using volcano plots of the levels of identified
metabolites in PDAC relative to NAT samples, considering the statistically significant
difference (p-value) and variable importance in the projection. As shown in Figure 4D
and 4E, the levels of the differential metabolites in PDAC were significantly different
from those in NAT in LC-MS and GC-MS profiling. Taken together, PDAC was
associated with significant changes in the metabolome from multiple profiling of
tumour and NAT samples.

***The altered metabolites and KEGG pathways in PDAC tissues compared with***
***NAT***

We then investigated the association of each annotated metabolite with the
PDAC group and identified 417 different metabolites in PDAC tissues compared with
NAT, including 138 elevated and 279 depleted metabolites from LC-MS profiling
(variable importance in projection (VIP) > 1 and Q value < 0.05) (Figure 4D). Using
GC-MS profiling, we identified 147 differential metabolites, of which 17 were
elevated and 130 were depleted (Figure 4E). Metabolite origin analysis revealed that

there were 149 differential metabolites associated with host and microbiota, including
4 host-specific metabolites, 43 bacterial metabolites and 102 bacteria–host
co-metabolites (Figure 4C). The abundance of 145 bacterial-related metabolites was
shown as heatmap (Supplementary figure 3). The depleted metabolites
glycerophosphocholine and 2-Lysophosphatidylcholine had the highest VIP scores,
whereas the most important enriched metabolite were oleamide, palmitoylcarnitine
and L-acetylcarnitine. Among the amino acid metabolites, beta-alanine, ergothioneine,
L-Isoleucine levels were significantly increased in PDAC, whereas other amino acids
and analogues, such as L-isoleucine, L-valine, L-aspartic acid, L-cysteine, L-serine
and L-glutamine were significantly reduced in PDAC samples. The roles of these
altered metabolites in PDAC need to be further studied, allowing for potential
correlation analysis based on metabolite–microbial interactions.

We conducted MPEA analysis on differential metabolites from the host,
microbiota, and shared by both. There were 3 and 82 metabolic pathways were related
to the bacteria and co-metabolism pathway database (Figure 4F). Among which, 3 and
32 metabolic pathways were identified significantly associated with PDAC,
correspondingly ($p < 0.05$)(Figure 4G). Based on origin-based function analysis, no
metabolic pathway was found specific related to host, while ascorbate and aldarate
metabolism, and folate biosynthesis were specific to the bacteria alone, and 32
metabolic pathways associated with amino acids, lipids, and sugars were shared by
both host and microbiota.

*The association between discriminative species and metabolites*

Our multi-omics data enabled us to identify dynamic interactions between
differential taxonomic and metabolic signatures. To dissect interactions between the
host and microbiota that might underlie features in PDAC, we assessed the
correlations between PDAC-related species, altered KO genes, and differentially
abundant metabolites originate from microbiota. And to explore more accurate
evidence of microbial enzyme–metabolite interactions, based on the reactions in the
KEGG database, we associated the altered metabolites with the discriminate KO gene
PDAC.

Broadly, in the spearman correlation analysis, we observed more significant
correlations in the NAT samples (Figure 5A), indicating an alteration of the
interaction between microbial species and metabolites. Among them, the
PDAC-enriched species *Cutibacterium acnes* and *Cutibacterium granulorum* were
broadly negatively correlated with differentially abundant metabolites in NAT
samples, while *Staphylococcus aureus* was mostly positively correlated (Figure 5A,
Supplementary figure 4). In PDAC tumour group, *Staphylococcus aureus* was
negatively correlated with beta-alanine and positively correlated with
K00823(4-aminobutyrate aminotransferase [EC:2.6.1.19]) (Figure 5B).

In the representative formula listed for chemical reactions involving
differential metabolite and bacterial enzyme genes, K00823 [EC:2.6.1.19] along with
its substrate beta-alanine and 2-oxoglutarate, were enriched in PDAC samples.

Glycerophosphocholine pathway metabolites sn-glycero-3-phosphocholine and
sn-glycero-3-phosphoethanolamine along with their product sn-glycerol 3-phosphate,
were depleted in PDAC samples, while K01126 [E3.1.4.46] enriched (Figure 5C).
The alteration of correlations between microbial species and metabolites might be
explained by the production of the species or the metabolites that favour the growth of
some species. Overall, though it remains to be further explored, our analysis
suggested a strong interaction between pancreatic microbial species, KO gene, and
metabolites in PDAC tumourigenesis and progression.

**Discussion**

This study reports an integrated analysis of the pancreatic microbiome,
metabolome, and predicted microbial KO genes in patients with PDAC. Owing to the
unavailability of normal, healthy pancreatic specimens, we used normal adjacent
tissue specimens. On the one hand, if this is a limit, the comparison of matched tumor
and NAT areas minimized interpatient confounding factors such as diet or lifestyle,
which are already known to significantly impact commensal microbiome
composition^(25, 26). Integrated analysis showed unique reactions in PDAC, providing
new mechanistic insights into the pathogenesis of the disease. In this study, the
α -diversity of pancreatic microbiota was found to be higher in normal adjacent tissues
than in tumor tissues, which is consistent with the results of a previous study⁽¹⁵⁾.
Many altered differential microbial species and metabolites have been identified
between PDAC and NAT samples, which may indicate a general mechanism for

PDAC.

Commensal opportunistic pathogens *Staphylococcus aureus*, *Cutibacterium*
*acnes* and *Cutibacterium granulosum* are enriched in PDAC tumor tissues, which is
consistent with the results of previous intratumor microbiome studies^(27, 28). Cavarretta
et al. described an increase in *Staphylococcus aureus* and *Cutibacterium acnes* in
prostate cancer. *Staphylococcus aureus* was found to colonize the tumor tissue of
breast cancer patients, and intratumor *Staphylococcus* significantly contributed to
tumor metastasis in animal experiments⁽²⁸⁾. Inflammation promotes the development
of tumors^(29, 30). *Cutibacterium acnes* has phosphatidylinositol and peptidoglycan in
the outer envelope, which contribute to the induction of an inflammatory response via
TLR-2 or TLR-4, thus playing a critical role in acne inflammation⁽³¹⁾. Davidsson et al.
found that *Cutibacterium acnes* contributes to an immunosuppressive environment in
prostate cancer by recruiting Tregs and by increasing the expression of
immunosuppressive mediators such as PD-L1, CCL17, and CCL⁽³²⁾. High
immunosuppression is an important characteristic of PDAC, and this result suggests
that *Cutibacterium acnes* may contribute to the formation of this immunosuppressive
microenvironment of PDAC. The abundance of gut commensal bacteria *Dialister*
*hominis*, a succinate consumer, was reduced in PDAC tumor tissues compared to
non-tumor tissues, which differs from previous findings in digestive tract cancer^(33, 34).
A meta-analysis based on six studies showed that the relative abundance of *Dialister*
was significantly higher in mucosal from gastric cancer patients than in control
samples⁽³³⁾. In addition, a high abundance of the genus *Dialister* in tumors, adjacent

tumors, and off-tumor areas was found to be associated with shorter overall survival
in colorectal cancer patients and works as an index for predicting the risk of colorectal
cancer recurrence and disease prognosis⁽³⁴⁾. This discrepancy in abundance may be
attributed to variances in the host organ being colonized or may stem from the finer
taxonomic resolution employed in this study, which extends to the species level.

In survival analysis, we found that patients with presence of *Ralstonia*
*pickettii_B* had significantly worse OS. *Ralstonia pickettii* is a non-fermenting
gram-negative commensal bacillus, and is an opportunistic pathogen that often causes
nosocomial infections(35). However, the effects of *Ralstonia pickettii* in
tumorigenesis has not been well studied. A study by Higuchi et al. observed that
*Ralstonia pickettii* was presented in almost all mesothelioma patients(36), while
causal studies are still needed to determine its potential and mechanisms in
tumorigenesis.

Following two untargeted approaches, our current metabolomic analysis
indicated a markedly larger number of decreasing lipid and lipid-like compounds than
increasing ones in the PDAC group, most of which could be assigned to classes of
fatty acyls and glycerophospholipids. Dysregulated lipid metabolism is now
recognized as a hallmark of many malignancies^(37, 38). High phosphorylcholine and
low glycerophosphorylcholine levels are consistently observed in aggressive cancers,
and an elevated phosphorylcholine/glycerophosphorylcholine ratio has also been
proposed as a biomarker of tumour progression⁽³⁹⁻⁴¹⁾. MPEA analysis further implied
that glycerophospholipid metabolism is a key pathway in PDAC pathogenesis. The

metabolites of the glycerophosphocholine pathway, namely
sn-glycero-3-phosphocholine and sn-glycero-3-phosphoethanolamine, along with
their resultant product sn-glycerol 3-phosphate, exhibited a depletion in samples of
PDAC. Concurrently, the enzyme K01126 [E3.1.4.46] demonstrated an enrichment in
PDAC samples.

Within the module of amino acid metabolism, MPEA analysis identified key
metabolic routes including alanine, aspartate, and glutamate metabolism, arginine
biosynthesis. Notably, there was an observed enrichment of beta-alanine, coupled
with a marked depletion of L-aspartic acid, L-glutamine, and L-serine in PDAC.
These alterations could potentially be attributed to variations in the pancreatic
microbiota and their associated enzymatic activities. Furthermore, the enzyme
K00823 [EC:2.6.1.19], along with its substrates beta-alanine and 2-oxoglutarate,
demonstrated a significant enrichment in PDAC. Beta-alanine was reported
suppresses tumour aggressiveness in vitro(42), simultaneously, the PDAC-enriched
*Staphylococcus aureus* exhibited a significant positive correlation with K00823, while
showing a marked negative correlation with the anti-tumour potential metabolites
beta-alanine. Suggesting that *Staphylococcus aureus* may influence the tumorigenesis
of PDAC through its involvement in amino acid metabolism.

By integrating multi-omics data, our study revealed a range of
microbiome-metabolite interactions. Association analysis indicated a markedly
reduced number of statistically significant correlations between microbial species and

metabolites within PDAC tumour samples, as opposed to the NAT samples, which
coincides with the decrease in bacterial α -diversity in PDAC progression.

Our study has several limitations. Although 2bRAD-M is very powerful in
characterizing microbiota at the species level and covering a comprehensive range of
species, it has a limitation in uncovering the full genetic content when compared to
metagenomic sequencing. We found that the M stage had a significant impact on
certain microbes, but the fact that only three M1 stage patients in our study cohort
required future exploration of advanced PDAC. Moreover, our study used pancreatic
tissues for metabolomic detection, and because of the complexity of the PDAC tumor
microenvironment, the detection results not only characterized the metabolic
alteration of PDAC tumour cells but also a picture of the entire microenvironment.

In conclusion, by leveraging multi-omics data, our study attempted to reveal
the common states of pancreatic microbiome dysbiosis and metabolome dysregulation
in patients with PDAC. We found that microbial species affect the tumorigenesis,
metastasis, and prognosis of PDAC and identified unique microbe–
enzyme-metabolite interaction. Although more mechanistic studies and clinical
validation are needed, our study highlights the need for further investigation of the
potential associations between pancreatic microbiota-derived omics signatures, which
may drive the clinical transformation of microbiome-derived strategies towards
therapy-targeted bacteria.

**Materials and methods**

*Study participants and sample collection*

A total of 105 patients diagnosed with PDAC who underwent surgery between
July 2016 and August 2022 at the Peking Union Medical College Hospital, Peking,
China, were enrolled for microbiome and untargeted metabolome analysis. Normal
adjacent tissues (NAT) were used as controls. All patients were diagnosed by
postoperative pathological examinations. Tissue samples were collected during
surgery and were stored at -80°C for microbiome sequencing and metabolic analysis.
Tumor stage was evaluated based on the tumor size, node, and metastasis (TNM)
staging system. Tumor differentiation was assessed from the same specimens using
the standard pathological grading scheme into well-differentiated, moderately
differentiated, or poorly differentiated based on the lowest differentiation grade
observed. Ethical approval for this study was obtained from the Human Research
Ethics Committee of Peking Union Medical College Hospital I-23PJ1417. The
requirement for consent was waived by the ethics committee due to retrospective
study.

*Microbiota analysis*

A detailed description of DNA extraction, amplification, sequencing processing,
and decontamination has been described in the supplementary materials. In brief,
genomic DNA of pancreatic tissues was extracted using a TIANamp Micro DNA Kit

(Tiangen, cat. #DP316). The 2bRAD-M library preparation method was primarily
based on the original protocol developed by Wang et al.⁽⁴³⁾, with slight modifications.
The PCR products were purified using a QIAquick PCR purification kit (Qiagen) and
then sequenced using the Illumina Nova PE150 platform. 2bRAD-M sequencing was
performed at Qingdao OE Biotech Co.,Ltd. (Qingdao, China). Reads with an N base
proportion greater than 8% and low-quality reads (with a base quality value below
Q30 constituting more than 20% of the total bases in a read) were removed during
sequence quality control. To identify the microbial species within each sample, the
sequenced 2bRAD tags underwent quality control and were subsequently mapped
against the 2bRAD marker database using a built-in Perl script. The relative
abundance of a specific species was determined by calculating the ratio of the number
of microbial individuals attributed to that species to the total number of individuals
from known species detectable within a given sample. The present study employed air
samples from laboratory and surgical environments as negative controls, utilizing a
combination of decontam, microDecon, and FEAST to effectively eliminate
background microorganisms from the experimental samples.

The relative abundance feature table was imported into R for further analysis.
The alpha diversity of each sample was evaluated using the Chao 1, Simpson, and
Shannon indices. Compositional differences between each pair of groups were
analyzed using permutational multivariate analysis of variance (PERMANOVA, 999
permutations), and distance matrix was constructed based on the Bray–Curtis distance

of the relative abundance of species. The compositional shift was further visualized
using principal coordinate analysis (PCoA) based on the same distance matrix.
Differential species analysis between PDAC and NAT groups was conducted using
the microbiome multivariable associations with a linear model (MaAsLin 2) and
Lefse. Multiple comparisons were adjusted using Benjamini–Hochberg method, $q <$
0.25.

*Survival analysis*

Overall survival (OS) includes death from any cause as events after the
perioperative period. For all endpoints, person-time is defined as time from surgery to
event or loss to follow-up (censored). Microbial species associated with OS were
assessed independently in the tumor samples, using repeated cross-validated
elastic-net penalized Cox proportional hazards regression. Species with low relative
abundance and prevalence were excluded, resulting in the inclusion of 26 species. We
conducted 10-fold cross-validated elastic-net penalized Cox regression using the
“cv.glmnet” function in the “glmnet” R package, with an α value of 0.5 to allow
groups of correlated predictors to be selected together. Clinical characteristics (age,
gender, BMI level, smoking, alcohol, family history, other malignancy, diabetes,
hyperlipemia, antibiotics usage, location, upregulation of CA19-9, differentiation,
perineural invasion (PNI), blood vessel invasion (BVI), T stage, N stage, M stage,
Stage, and neoadjuvant therapy) were included in the model. The effects of species
and clinical characteristics on OS were investigated using Kaplan–Meier survival

curves and compared using log-rank test.

*Metabolome data analysis*

A detailed description of sample preparation, experiment condition, and data
processing for LC-MS and GC-MS untargeted metabolome analysis has been
described in the supplementary materials. In brief, the original LC-MS data were
processed using Progenesis QI V2.3 (Nonlinear, Dynamics, Newcastle, UK) for
baseline filtering, peak identification, integration, retention time correction, peak
alignment, and normalization. The GC/MS raw data were obtained from. The D
format is transferred to. abf format using the software Analysis Base File Converter
for the quick retrieval of data. The data were then imported into MS-DIAL software,
which performs peak detection, peak identification, MS2Dec deconvolution,
characterization, peak alignment, wave filtering, and missing value interpolation.
After the data were normalized, redundancy removal and peak merging were
performed to obtain the data matrix.

The matrix was imported into R for analysis. OPLS-DA was used to identify
differentiating metabolites between the groups. To mitigate overfitting, a 7-fold
cross-validation and 200 Response Permutation Testing (RPT) were performed to
evaluate the model's quality. Variable Importance of Projection (VIP) values derived
from the OPLS-DA model were used to rank the overall contribution of each variable
to group discrimination. Subsequently, a two-tailed Student's t-test was conducted to
verify the statistical significance of the identified metabolites differentiating between

the groups. Differential metabolites were selected based on VIP values greater than
1.0 and p-values less than 0.05. Metabolites origin was analysed using Metorigin.

*Analysis of microbiome-metabolites interactions*

Spearman correlation analysis was performed using the 'psych' package in R to
investigate the associations between differential microbial species and differential
metabolites. Only associations with a correlation coefficient (R) greater than 0.3 and
an $p < 0.05$ were considered significant. The resulting associations were visualized
using a heatmap.

*Statistical analyses*

All pairwise comparisons were performed using a two-sided Wilcoxon rank-sum
test (Mann-Whitney U test). Dissimilarity tests among groups (PERMANOVA) were
conducted on Euclidean distance for metabolites and Bray-Curtis distance for
bacteria, with 999 permutations in the R package vegan. Multiple comparisons were
adjusted using Benjamini–Hochberg method. All statistical analyses were performed
using R, version 4.2.1.

*Data availability*

The datasets used and analysed during the current study are included in this
published article and its supplementary information files. No unique code was
generated in this study. The code and raw sequencing data for this study is available

upon request from corresponding author Lei You, florayo@163.com.

**Ethics approval**

Ethical approval for this study was obtained from the Human Research Ethics
Committee of Peking Union Medical College Hospital I-23PJ1417. The requirement
for consent was waived by the ethics committee.

**Acknowledgments**

The authors would like to thank Professor Taiping Zhang for his efforts in
sample collection. Wenbo Kou of OE Biotechnology Company (Qingdao, China) for
his technical support in removing contamination from the sequencing data. This study
was funded by the National High Level Hospital Clinical Research Funding
(2022-PUMCH-D-001), Chinese Academy of Medical Sciences (CAMS) Innovation
Fund for Medical Sciences (CIFMS, 2021-I2M-1-002), National Multidisciplinary
Cooperative Diagnosis and Treatment Capacity Building Project for Major Diseases
(NSFC, 81970763), Nonprofit Central Research Institute Fund of CAMS
(PT201832014), National Natural Science Foundation of China (NSFC, 82103016)
and (NSFC, 62133006). The funders played no role in study design, data collection,
analysis and interpretation of data, or the writing of this manuscript.

**Conflict of interests**

All authors declare no financial or non-financial competing interests.

**Contributions**

LY and YPZ supervised the study. TPZ provided samples from patients. GY, JR,
XW, and FHZ collected samples and provided clinical information. YF and BR
performed the bioinformatics analyses. RLX, MZG, JLB, XYL, and SH have
provided helpful comments. YF, BR, and XHL interpreted the results. YF and BR
wrote the manuscript. All authors have read and approved the final manuscript.

**Reference**

- 1. Mizrahi JD, Surana R, Valle JW, Shroff RT. 2020. Pancreatic cancer. *Lancet*
395:2008-2020.
- 2. Rahib L, Wehner MR, Matrisian LM, Nead KT. 2021. Estimated Projection of US
Cancer Incidence and Death to 2040. *JAMA Netw Open* 4:e214708.
- 3. Siegel RL, Miller KD, Wagle NS, Jemal A. 2023. Cancer statistics, 2023. *CA Cancer J*
*Clin* 73:17-48.
- 4. Nevala-Plagemann C, Hidalgo M, Garrido-Laguna I. 2020. From state-of-the-art
treatments to novel therapies for advanced-stage pancreatic cancer. *Nat Rev Clin*
*Oncol* 17:108-123.
- 5. Overbeek KA, Levink IJM, Koopmann BDM, Harinck F, Konings I, Ausems M, Wagner
537 A, Fockens P, van Eijck CH, Groot Koerkamp B, Busch ORC, Besselink MG,
Bastiaansen BAJ, van Driel L, Erler NS, Vleggaar FP, Poley JW, Cahen DL, van Hooft
JE, Bruno MJ, Dutch Familial Pancreatic Cancer Surveillance Study G. 2022.

- Long-term yield of pancreatic cancer surveillance in high-risk individuals. *Gut*
71:1152-1160.
- 6. Hanahan D. 2022. Hallmarks of Cancer: New Dimensions. *Cancer Discovery*
12:31-46.
- 7. McQuade JL, Daniel CR, Helmink BA, Wargo JA. 2019. Modulating the microbiome to
improve therapeutic response in cancer. *Lancet Oncol* 20:e77-e91.
- 8. Human Microbiome Project C. 2012. Structure, function and diversity of the healthy
human microbiome. *Nature* 486:207-14.
- 9. de Martel C, Ferlay J, Franceschi S, Vignat J, Bray F, Forman D, Plummer M. 2012.
Global burden of cancers attributable to infections in 2008: a review and synthetic
analysis. *Lancet Oncol* 13:607-15.
- 10. Nejman D, Livyatan I, Fuks G, Gavert N, Zwang Y, Geller LT, Rotter-Maskowitz A,
Weiser R, Mallel G, Gigi E, Meltser A, Douglas GM, Kamer I, Gopalakrishnan V,
Dadosh T, Levin-Zaidman S, Avnet S, Atlan T, Cooper ZA, Arora R, Cogdill AP, Khan
MAW, Ologun G, Bussi Y, Weinberger A, Lotan-Pompan M, Golani O, Perry G, Rokah
555 M, Bahar-Shany K, Rozeman EA, Blank CU, Ronai A, Shaoul R, Amit A, Dorfman T,
Kremer R, Cohen ZR, Harnof S, Siegal T, Yehuda-Shnaidman E, Gal-Yam EN,
Shapira H, Baldini N, Langille MGI, Ben-Nun A, Kaufman B, Nissan A, Golan T,
Dadiani M, et al. 2020. The human tumor microbiome is composed of tumor
type-specific intracellular bacteria. *Science* 368:973-980.
- 11. Geller LT, Barzily-Rokni M, Danino T, Jonas OH, Shental N, Nejman D, Gavert N,
Zwang Y, Cooper ZA, Shee K, Thaiss CA, Reuben A, Livny J, Avraham R, Frederick

DT, Ligorio M, Chatman K, Johnston SE, Mosher CM, Brandis A, Fuks G, Gurbatri C,
Gopalakrishnan V, Kim M, Hurd MW, Katz M, Fleming J, Maitra A, Smith DA, Skalak
564 M, Bu J, Michaud M, Trauger SA, Barshack I, Golan T, Sandbank J, Flaherty KT,
Mandinova A, Garrett WS, Thayer SP, Ferrone CR, Huttenhower C, Bhatia SN,
Gevers D, Wargo JA, Golub TR, Straussman R. 2017. Potential role of intratumor
bacteria in mediating tumor resistance to the chemotherapeutic drug gemcitabine.
Science 357:1156-1160.

12. Thomas RM, Gharaibeh RZ, Gauthier J, Beveridge M, Pope JL, Gujjarro MV, Yu Q,
He Z, Ohland C, Newsome R, Trevino J, Hughes SJ, Reinhard M, Winglee K, Fodor
AA, Zajac-Kaye M, Jobin C. 2018. Intestinal microbiota enhances pancreatic
carcinogenesis in preclinical models. Carcinogenesis 39:1068-1078.

13. Pushalkar S, Hundeyin M, Daley D, Zambirinis CP, Kurz E, Mishra A, Mohan N, Aykut
B, Usyk M, Torres LE, Werba G, Zhang K, Guo Y, Li Q, Akkad N, Lall S, Wadowski B,
Gutierrez J, Kochen Rossi JA, Herzog JW, Diskin B, Torres-Hernandez A, Leinwand J,
Wang W, Taunk PS, Savadkar S, Janal M, Saxena A, Li X, Cohen D, Sartor RB,
Saxena D, Miller G. 2018. The Pancreatic Cancer Microbiome Promotes Oncogenesis
by Induction of Innate and Adaptive Immune Suppression. Cancer Discov 8:403-416.

14. Dickson I. 2018. Microbiome promotes pancreatic cancer. Nat Rev Gastroenterol
Hepatol 15:328.

15. Del Castillo E, Meier R, Chung M, Koestler DC, Chen T, Paster BJ, Charpentier KP,
Kelsey KT, Izard J, Michaud DS. 2019. The Microbiomes of Pancreatic and
Duodenum Tissue Overlap and Are Highly Subject Specific but Differ between

Pancreatic Cancer and Noncancer Subjects. *Cancer Epidemiol Biomarkers Prev*
28:370-383.

16. Riquelme E, Zhang Y, Zhang L, Montiel M, Zoltan M, Dong W, Quesada P, Sahin I,
Chandra V, San Lucas A, Scheet P, Xu H, Hanash SM, Feng L, Burks JK, Do KA,
Peterson CB, Nejman D, Tzeng CD, Kim MP, Sears CL, Ajami N, Petrosino J, Wood
LD, Maitra A, Straussman R, Katz M, White JR, Jenq R, Wargo J, McAllister F. 2019.

[revised manuscript text omitted]

(E) age. Log-rank test used to generate p-values.

Figure 4. Metabolome profile and function changes in PDAC. (A) and (B) OPLS-DA

showed that PDAC and NAT samples were separated into two distinct clusters. (D, E)

Volcano plot demonstrated metabolites changes between 90 PDAC and 98 NAT

samples. The x-axis indicates log₂-transformed fold change of metabolite

abundances, and the y-axis denotes log₁₀-transformed Q values (p value adjusted

using the tail area-based FDR). The horizontal lines represent $q < 0.05$. (C) and (F)

Venn plot showed results of metabolites origin analysis. (G) The functions of

discriminate metabolites originated from microbiota, metabolic pathways were

studied using the KEGG database, and enriched pathways were displayed by a bubble

plot. The size of bubble represents the number of metabolites detected in the KEGG

pathway.

Figure 5. Integrated analysis of multi-omics in NAT and PDAC. (A, B) The network

revealed representatively significant and suggestive associations ($p < 0.05$, Spearman

analysis) among differentially abundant taxa, metabolites and KO genes (A) NAT

group, and (B) PDAC group. Lines connecting nodes indicate positive (red) or

negative (blue) correlations. (C) Representative metabolites-KO gene reactions.

Representative enzymes, metabolites appearing in the existing PDAC metabolite-

enzyme reactions are shown in the formula listed. Each boxplot in a reaction

represents a compound or a KO gene (two-side Wilcoxon rank-sum test). Boxplots

show relative metabolite concentrations or KO gene abundances averaged over
samples within each group and are coloured according to the group.

Table 1. Characteristics of PDAC patients enrolled.

Characteristics	Sample Information Statistics
n (patient)	105
Gender = M (%)	63 (60.0)
Age (mean (SD))	61.57 (8.18)
Family_history = yes (%)	20 (19.0)
Pancreatitis = yes (%)	31 (30.1)
Other_malignancy = yes (%)	8 (7.6)
BPD = yes (%)	17 (16.2)
EUS_FNA = yes (%)	11 (10.5)
Weight_loss (mean (SD))	4.19 (4.82)
BMI (mean (SD))	24.76 (3.20)
Smoking (%)	
never	58 (55.2)
ever	12 (11.4)
current	35 (33.3)
Diabetes (%)	
new-onset	14 (13.3)
no	66 (62.9)
yes	25 (23.8)
Alcohol (%)	
never	65 (61.9)
ever	2 (1.9)
current	38 (36.2)
Hyperlipidemia (%)	
dyslipidemia	14 (13.3)
no	58 (55.2)
yes	33 (31.4)
Biliary_disease = yes (%)	62 (59.0)
Antibiotics = yes (%)	10 (9.5)
Location = tail (%)	44 (41.9)
CA19_9_upregulate = yes (%)	85 (81.0)
Differentiation (%)	
poor	45 (42.9)
moderate	43 (41.0)
well	17 (16.1)
PN I = yes (%)	74 (70.5)
BVI = yes (%)	40 (38.1)
T_stage (%)	
T1	27 (25.7)
T2	47 (44.8)
T3	28 (26.7)
T4	3 (2.9)
N_stage (%)	
N0	51 (48.6)
N1	37 (35.2)
N2	17 (16.2)
M_stage = M1 (%)	3 (2.9)
Stage (%)	
IA	15 (14.3)
IB	21 (20.0)
IIA	13 (12.4)
IIB	34 (32.4)
III	19 (18.1)
IV	3 (2.9)
Neoadjuvant_therapy = yes (%)	4 (3.8)

**Integrated Analysis of Microbiome and Metabolome Reveals**

**Signatures in PDAC Tumorigenesis and Prognosis**

Yuan Fang¹²³⁴, Xiaohong Liu¹²³⁴, Jie Ren¹²³⁴, Xing Wang¹²³⁴, Feihan
Zhou¹²³⁴, Shi Huang⁵, Lei You^{1234*} and Yupei Zhao^{1234*}

**Author affiliations and addresses:**

1.Department of General Surgery, Peking Union Medical College Hospital, Peking
Union Medical College, Chinese Academy of Medical Sciences, 100730, Beijing,
China.

2.Key Laboratory of Research in Pancreatic Tumor, Chinese Academy of Medical
Sciences, 100730, Beijing, China.

3.National Science and Technology Key Infrastructure on Translational Medicine in
Peking Union Medical College Hospital, 100730, Beijing, China.

4.State Key Laboratory of Complex Severe and Rare Diseases, Peking Union Medical
College Hospital, Chinese Academy of Medical Sciences and Peking Union Medical
College, 100730, Beijing, China.

5.Faculty of Dentistry, The University of Hong Kong, 999077, Hong Kong SAR,
China

**E-mail addresses:**

Yuan Fang, 985311609@qq.com

Xiaohong Liu, Liu_xh1001@163.com

Jie Ren, 18813126010@163.com

Xing Wang, drwang99@163.com

Feihan Zhou, 15261588335@163.com

Shi Huang, shihuang@hku.hk

***Authors for correspondence:**

Lei You, florayo@163.com; Tel: +86-10-69156007. No. 1 Shuaifuyuan, Wangfujing

Street, Beijing 100730, China

Yupei Zhao, E-mail: zhao8028@263.net; Tel: +86-10-69156007; Fax:

+86-10-65124875; No. 1 Shuaifuyuan, Wangfujing Street, Beijing 100730, China

**Abstract**

Pancreatic cancer, predominantly pancreatic ductal adenocarcinoma (PDAC), is one of the
most malignant tumors of the digestive system. Emerging evidence suggests the involvement
of the microbiome and metabolic substances in the development of PDAC, yet the results
remain contradictory. This study aims to identify the alterations and relationships in
intratumoral microbiome and metabolites in PDAC. We collected matched tumor and normal
adjacent tissue (NAT) samples from 105 PDAC patients and performed a 6-year follow-up.
2bRAD-M sequencing, untargeted liquid chromatography-tandem mass spectrometry
(LC-MS), and untargeted gas chromatography-mass spectrometry (GC-MS) were performed.
Compared with NATs, microbial alpha diversity decreased in PDAC tumor. The relative
abundance of *Staphylococcus aureus*, *Cutibacterium acnes* and *Cutibacterium granulosum*
were higher in PDAC tumor after adjusting for confounding factors BMI and M stage, and the
presence of *Ralstonia pickettii_B* was found associated to a worse overall survival.
Metabolomic analysis revealed distinctive differences in composition between PDAC and
NAT, with 553 discriminative metabolites identified. Differential metabolites were revealed
originated from the microbiota and showed significant interactions with shifts of bacterial
species through KO genes. These findings suggest that the PDAC microenvironment harbors
unique microbial-derived enzymatic reactions, potentially influencing the occurrence and
development of PDAC by modulating the levels of glycerol-3-phosphate, succinate,
carbonate, and beta-alanine.

**Importance**

We conducted a large sample size pancreatic adenocarcinoma microbiome study
using a novel microbiome sequencing method and two metabolomic assays. Two
significant outcomes of our analysis are (a) Commensal opportunistic pathogens
*Staphylococcus aureus*, *Cutibacterium acnes* and *Cutibacterium granulosum* were
enriched in PDAC tumours compared with normal adjacent tissues, and worse overall
survival was found related to presence of *Ralstonia pickettii*_B. (b). Microbial species
affect the tumorigenesis, metastasis, and prognosis of PDAC via unique microbe–
enzyme-metabolite interaction. Thus, our study highlights the need for further
investigation of the potential associations between pancreatic microbiota-derived
omics signatures, which may drive the clinical transformation of microbiome-derived
strategies towards therapy-targeted bacteria.

**Keywords:** PDAC; pancreatic ductal adenocarcinoma; microbial metabolism; microbiota;
metabolome; carcinogenesis

**Introduction**

Pancreatic ductal adenocarcinoma (PDAC) remains a highly fatal malignancy,
with a 5-year overall survival of 12%⁽¹⁻³⁾. Therapeutic methods for PDAC remain
limited⁽⁴⁾. Surgery combined with adjuvant chemotherapy remains the only curative
therapeutic option, but more than 60% of patients are diagnosed with unresectable

disease⁽⁵⁾. The pathogenesis of PDAC is complex, and genetic alterations in PDAC
fail to explain carcinogenesis alone, which leaves environmental factors including the
microbiota emerging as potential mediators of PDAC carcinogenesis. Thus, as an
important hallmark of cancer⁽⁶⁾, the role of polymorphic microbiomes in cancer
remains to be explored, and there is a pressing need to identify microorganisms that
might explain the differences between PDAC tumours and normal pancreatic tissue so
that new concepts can be developed for future therapies⁽⁷⁾.

Numerous studies have shown that microorganisms play a critical role in the
carcinogenesis^(8, 9). Intratumoral bacteria have been observed in various tumours,
including PDAC⁽¹⁰⁾, which is associated with PDAC carcinogenesis, progression, and
poor prognosis via a complex mechanism. Geller et al. initially proposed a connection
between intratumoural microbiota and PDAC⁽¹¹⁾. The prevalence of intratumoural
bacteria in PDAC tissues was significantly higher than that in normal pancreatic
tissues. The presence of microbiota in the pancreas of both healthy and cancerous
subjects was also confirmed in a study by Thomas et al.⁽¹²⁾. In a study involving 12
PDAC patients, Pushalka et al. found higher bacterial biomass in PDAC tumours than
in normal pancreatic tissue^(13, 14). Alpha diversity was reported to be slightly higher in
healthy controls versus in patients⁽¹⁵⁾. The most common class identified in the PDAC
intratumor microbiome is Gammaproteobacteria, with the dominant genus
*Pseudomonas*^(11, 13), which carries long-form cytidine deaminase (CDDL) that
metabolizes the chemotherapeutic drug gemcitabine (2',2'-difluorodeoxycytidine) into
its inactive form (2',2'-difluorodeoxyuridine)⁽¹¹⁾. Riquelme et al. investigated the

impact of tumour microbiota on PDAC patient survival⁽¹⁶⁾. Patients with long-term
survival exhibited higher alpha diversity and enrichment for *Pseudoxanthomonas*,
*Saccharopolyspora*, and *Streptomyces*. Guo et al. revealed that *Acinetobacter*,
*Pseudomonas*, and *Sphingopyxis*, as the intratumoural microbiota of basal-like
PDAC, were associated with worse prognosis by inducing inflammation⁽¹⁷⁾. Taken
together, these data support that distinctive profiles of tumour microbiota may
underlie PDAC heterogeneity, and comprehensive characterization of the PDAC
intratumoral microbiome may be an essential step in unraveling the effects of bacteria
on PDAC tumourigenesis and prognosis. Despite these developments, the clinical
significance of the intratumoral microbiome in PDAC is still poorly understood.
Previous comparative PDAC-healthy control studies generally constrained by
limitations of small number of samples and vague taxonomy classification, thus, large
sample size studies are urgently required. Therefore, it is imperative to further
investigate the profiles of the PDAC tumour microbiome and clarify its clinical
significance and prognostic value.

Microbiota-derived metabolites are important natural products that establish a
strong connection between the microbiome and cancer⁽¹⁸⁾. For example, microbial
byproducts can actively contribute to carcinogenesis. Secondary metabolites,
including lithocholic acid and deoxycholic acid^(19, 20), as well as catabolites, such as
acetate and butyrate^(21, 22), play a crucial role in enhancing either epithelial-
mesenchymal transition or cell proliferation in several models of cancer⁽²³⁾.
Metabolomic comparisons of human PDAC tumour tissue and normal adjacent tissue

(NAT) revealed that tumour tissues exhibit lower levels of glucose, upper glycolytic
intermediates, creatine phosphate, and the amino acids glutamine and serine, which
are the major metabolic substrates⁽²⁴⁾. However, evidence of the involvement of
microbiome-derived metabolites in PDAC carcinogenesis is limited.

In summary, to advance our understanding of the microbiome and
metabolome characteristics associated with PDAC, and to elucidate the intricate role
of their interaction in PDAC carcinogenesis and prognosis, we conducted
comprehensive analyses of the microbiome and metabolome of surgically excised
PDAC tumor and its matched NATs from a large scale of 105 patients based on
2bRAD-M sequencing, untargeted liquid chromatography-tandem mass spectrometry
(LC-MS), and untargeted gas chromatography-mass spectrometry (GC-MS). Our
study provided data support for subsequent studies on PDAC, thereby expanding the
perspective in this field.

**Results**

*Participant characteristics*

We collected 208 tissue samples (103 matched PDAC and NATs, plus unpaired
2 NATs) from 105 patients. The demographic, clinical, and pathological
characteristics of the patients are shown in Table 1. The average age was 61.57 ± 8.18 ,
and 63 patients (60.0%) were male. Diabetes was present in 39 (37.1%) patients, and
among them 14 (13.3%) were new-onset diabetes. The number of patients with TNM

stages I, II, III and IV were 36 (34.3%), 47 (32.4%), 19 (18.1%) and 3 (2.9%),
respectively. Tumour differentiation was well (16.1%), moderate (41.0%) and poor
(42.9%).

*PDAC and NAT microbiome differ from global scale*

Negative controls were used in the DNA extraction and PCR amplification
processes to remove the interference of contaminating microorganisms introduced
during the experimental manipulation. Using a combination of decontam,
microDecon, and FEAST, background microorganisms in the environmental samples
of operating room and laboratory were deducted based on the negative controls. After
decontamination, 1920 species were identified. A broad overview of our taxonomic
data from the 105 subjects is provided in Supplementary Figure 1.

For α -diversity, significant decrease was observed in the in PDAC (Figure 1A;
$p = 0.0017, 0.00012, \text{ and } 0.00089$, respectively). Given the significant differences in
α -diversity between NAT and PDAC tissues, we explored the compositional diversity
of the microbiota in these two groups. A Venn diagram of the microbiota composition
showed that NAT had a greater variety of microorganisms (Figure 1B). At the species
level, 512 microorganisms were shared by the NAT and PDAC groups. The
Bray-Curtis distance was used to perform principal coordinate analysis (PCoA) of the
samples (Figure 1C). The results of the PERMANOVA test indicated a statistically
significant difference in the β -diversity between PDAC and NAT ($R^2 = 0.011$; $p =$
0.01).

Next, we investigated the potentially relevant influence factors for microbiome
alterations. PERMANOVA was used to explore the associations between variations in
the pancreatic microbiota and host characteristics. Given a false discovery rate (FDR)
of 5%, three parameters were significantly associated with microbial variations
derived from Bray–Curtis distances calculated on species level (Figure 1C). Group, M
stage, and BMI level were explanatory factors, which is consistent with previous
research.

***Taxonomic signatures of microbiota in PDAC tumor***

Next, we attempted to identify PDAC-associated taxa using multivariable
microbiome associations with a linear model (MaAsLin 2) to control for confounding
factors⁽²⁵⁾. The model included group as fixed effects and BMI, M stage as random
effect. Thirteen species between the PDAC and NAT groups were identified as
differentially abundant bacterial species (Figure 2A and 2C). Samples from the PDAC
group were enriched in *Staphylococcus aureus*, *Cutibacterium acnes* and
*Cutibacterium granulosum*, whereas depleted in *Sphingomonas aquatilis*, *BACL27*
*sp014190055*, *QWOQ01 sp003669585*, *Limnohabitans_A sp005789685*,
*Mycobacterium koreense*, *Mycobacterium intermedium*, *UBA953 sp002293125*,
*Bacillus_A bombysepticus*, *Pelomonas sp003963075* and *Dialister hominis*
(Supplementary table S2). After adjusting for M stage, the associations were still
significant. To compare our results with previous studies, we performed difference
analysis at the genus level, the results were broadly consistent with the species level.

Besides, *Bifidobacterium* was slightly increased in PDAC, while *Dietzia* and
*Streptococcus* were depleted (Figure 2B and 2D).

Based on the results of PERMANOVA (Figure 1C), we analysed the bacterial
alterations related to M stage. The pathogens *Pseudomonas fulva*, *Dietzia maris*,
*Massilia timonae* and *Brevundimonas diminuta* were positively associated with the
M1 stage, whereas *Pseudomonas_E sp900187635* was depleted in the M1 stage
(Figure 3A). In a subset containing only PDAC samples, these results still hold.
However, it is important to note that only three patients were diagnosed with stage
M1, rendering this result incidental.

To investigate the function of the intratumoural microbiota in PDAC, we
predicted the biological functions of the bacteria utilizing PICRUSt2. We identified
7301 KO genes altogether. Using the MaAsLin 2 analysis, a total of 1079 KO genes
were found to be differentially expressed between the two groups after adjusting for
BMI (Supplementary data).

***Microbial species related to overall survival***

Analysis of the relationship between microbial species and overall survival is
performed only in PDAC tumour samples. Among them, one patient with
perioperative cardiac death and four patients with uncertain time of death were
excluded, a total of 98 PDAC patients were included in the survival analysis, resulting
in a median of 15 months of follow-up (range 1–71 months).

In 10-fold cross-validated elastic-net Cox regression models for OS, we found

species *Ralstonia pickettii_B* and age were associated with OS. Based on these results,
we then tested the relationship between *Ralstonia pickettii_B* and OS by stratifying
the patients in two groups based on presence of *Ralstonia pickettii_B*. As expected,
we found that patients colonized with *Ralstonia pickettii_B* had significantly worse
OS (median OS: 17 months) than those *Ralstonia pickettii_B* negative ones (median
OS: 37 months) using Kaplan-Meier curve tested by log-rank ($p = 0.045$) (Figure 3D).
Given that PDAC is a disease in which risk increases with age, we stratified the
patients into two groups by age 65. A median OS of 36 months was obtained for
middle-aged group and 17 months for elder patients group ($p = 0.0047$) (Figure 3E).
Subgroup analyses of age and colonization of *Ralstonia pickettii_B* found that only
the *Ralstonia pickettii_B*-negative group had a difference in OS between middle-aged
and older adults, but the older group was too under-represented. Our follow-up is
ongoing and we believe that the existing results will be updated as the study
continues. Our findings indicate that the presence of *Ralstonia pickettii_B* in tumour
could serve as a predictor of survival outcome in resected PDAC patients, further
elaborated the potential of the microbiome composition in mediating PDAC
progression.

***Untargeted metabolomics profiling revealed significantly altered metabolites***

To investigate the interactions between the pancreatic microbiota and
host-microbe co-metabolism, untargeted metabolomic profiling was performed on a
subset of 98 NAT and 90 PDAC samples. We conducted both LC-MS and GC-MS to

make our assay as comprehensive as possible. A total of 6375 metabolites were
quantified from tissue samples using LC-MS and 481 were quantified using GC-MS.

Orthogonal partial least squares-discriminant analysis (OPLS-DA) (Figure 4A
and 4B) showed that there were differences in the tissue metabolite profiles between
PDAC and NAT groups, indicating a tumor-metabolite shift in PDAC carcinogenesis.
The ability of the OPLS-DA model was tested during a seven-cross validation through
200 random permutation tests. The intercepts of goodness-of-fit (R²) and
goodness-of-prediction (Q²) illustrate that the OPLS-DA model is reliable and does
not overfit. We plotted fold changes using volcano plots of the levels of identified
metabolites in PDAC relative to NAT samples, considering the statistically significant
difference (p-value) and variable importance in the projection. As shown in Figure 4D
and 4E, the levels of the differential metabolites in PDAC were significantly different
from those in NAT in LC-MS and GC-MS profiling. Taken together, PDAC was
associated with significant changes in the metabolome from multiple profiling of
tumour and NAT samples.

***The altered metabolites and KEGG pathways in PDAC tissues compared with***
***NAT***

We then investigated the association of each annotated metabolite with the
PDAC group and identified 417 different metabolites in PDAC tissues compared with
NAT, including 138 elevated and 279 depleted metabolites from LC-MS profiling
(variable importance in projection (VIP) > 1 and Q value < 0.05) (Figure 4D). Using

GC-MS profiling, we identified 147 differential metabolites, of which 17 were
elevated and 130 were depleted (Figure 4E). Metabolite origin analysis revealed that
there were 149 differential metabolites associated with host and microbiota, including
4 host-specific metabolites, 43 bacterial metabolites and 102 bacteria–host
co-metabolites (Figure 4C). The abundance of 145 bacterial-related metabolites was
shown as heatmap (Supplementary figure 4). The depleted metabolites
glycerophosphocholine and 2-Lysophosphatidylcholine had the highest VIP scores,
whereas the most important enriched metabolite were oleamide, palmitoylcarnitine
and L-acetylcarnitine. Among the amino acid metabolites, beta-alanine, ergothioneine,
L-Isoleucine levels were significantly increased in PDAC, whereas other amino acids
and analogues, such as L-isoleucine, L-valine, L-aspartic acid, L-cysteine, L-serine
and L-glutamine were significantly reduced in PDAC samples. The roles of these
altered metabolites in PDAC need to be further studied, allowing for potential
correlation analysis based on metabolite–microbial interactions.

We conducted metabolite pathway enrichment analysis (MPEA) analysis on
differential metabolites from the host, microbiota, and shared by both. There were 3
and 82 metabolic pathways were related to the bacteria and co-metabolism pathway,
respectively (Figure 4F). Among which, 3 and 32 metabolic pathways were identified
significantly associated with PDAC, correspondingly ($p < 0.05$) (Figure 4G). Based
on origin-based function analysis, no metabolic pathway was found specific related to
host, while ascorbate and aldarate metabolism, and folate biosynthesis were specific

to the bacteria alone, and 32 metabolic pathways associated with amino acids, lipids,
and sugars were shared by both host and microbiota.

***The association between discriminative species and metabolites***

Our multi-omics data enabled us to identify dynamic interactions between
differential taxonomic and metabolic signatures. To dissect interactions between the
host and microbiota that might underlie features in PDAC, we assessed the
correlations between PDAC-related species, altered KO genes, and differentially
abundant metabolites originate from microbiota in PDAC and NAT, respectively. And
to explore more accurate evidence of microbial enzyme–metabolite interactions,
based on the reactions in the KEGG database, we associated the altered metabolites
with the discriminate KO gene which significantly correlated to both species and
metabolites.

Broadly, in the spearman correlation analysis, we observed more significant
correlations in the NAT samples (Supplementary Figure 5), a total of 46 differential
metabolites were found to be significantly correlated with 8 differential species, and
302 KO genes were identified to be concurrently associated with both, resulting in a
cumulative total of 2523 correlations (Supplementary Figure 7A). Among them, the
PDAC-enriched species *Cutibacterium acnes* and *Cutibacterium granulosum* were
broadly negatively correlated with differentially abundant metabolites in NAT
samples (Supplementary figure 5). While in PDAC samples, 397 KO genes were
found to form 1260 correlations with 16 differential metabolites and 5 differential

species (Supplementary Figure 7B). *Cutibacterium granulosum* was only negatively
correlated with D-ornithine, and *Cutibacterium acnes* was only negatively correlated
with EPA (d5), meanwhile, *Cutibacterium acnes* and *Cutibacterium granulosum*
formed new positive correlations with carbonate, and *Cutibacterium acnes* formed a
new negative correlation with L-serine (Supplementary Figure 6).

In the representative formula listed for chemical reactions involving
differential metabolite and bacterial enzyme genes (Figure 5C and 5D), the levels of
substrate sn-glycero-3-phosphoethanolamine (GPE) and its product
glycero-3-phosphate (G3P) were significantly reduced in PDAC samples, while the
enzymes metabolizing them, namely glpQ and ugpQ, were elevated. Similarly, pcrB,
which catalyzes the conversion of O-succinyl-L-homoserine and hydrogen sulfide to
L-homocysteine and succinate, respectively, was upregulated in PDAC samples,
while the levels of its reaction product succinate decreased. In other identified
chemical reactions, enzymes tsaC, rimN, and SUA5 responsible for carbonate
decomposition were significantly enriched in carbonate content. Additionally,
beta-alanine and 2-oxoglutarate were significantly enriched compared to adjacent
control tissues, and the enzyme gene puuE catalyzing their reaction to produce
glutamate was also significantly enriched. In summary, though it remains to be further
explored, our analysis indicates that the PDAC microenvironment exhibits distinct
enzymatic reactions and metabolic processes originating from microorganisms.

**Discussion**

This study reports an integrated analysis of the pancreatic microbiome,
metabolome, and predicted microbial KO genes in patients with PDAC. Owing to the
unavailability of normal, healthy pancreatic specimens, we used normal adjacent
tissue specimens. On the one hand, if this is a limit, the comparison of matched tumor
and NAT areas minimized interpatient confounding factors such as diet or lifestyle,
which are already known to significantly impact commensal microbiome
composition^(26, 27). Integrated analysis showed unique reactions in PDAC, providing
new mechanistic insights into the pathogenesis of the disease. In this study, the
α -diversity of pancreatic microbiota was found to be higher in normal adjacent tissues
than in tumor tissues, which is consistent with the results of a previous study⁽¹⁵⁾.
Many altered differential microbial species and metabolites have been identified
between PDAC and NAT samples, which may indicate a general mechanism for
PDAC.

Commensal opportunistic pathogens *Staphylococcus aureus*, *Cutibacterium*
*acnes* and *Cutibacterium granulosum* are enriched in PDAC tumor tissues, which is
consistent with the results of previous intratumor microbiome studies^(28, 29). Cavarretta

[revised manuscript text omitted]

Tumor stage was evaluated based on the tumor size, node, and metastasis (TNM)
staging system. Tumor differentiation was assessed from the same specimens using
the standard pathological grading scheme into well-differentiated, moderately
differentiated, or poorly differentiated based on the lowest differentiation grade
observed. Ethical approval for this study was obtained from the Human Research
Ethics Committee of Peking Union Medical College Hospital I-23PJ1417. The
requirement for consent was waived by the ethics committee due to retrospective
study.

***Microbiota analysis***

A detailed description of DNA extraction, amplification, sequencing processing,
and decontamination has been described in the Supplementary materials. In brief,
genomic DNA of pancreatic tissues was extracted using a TIANamp Micro DNA Kit
(Tiangen, cat. #DP316). The 2bRAD-M library preparation method was primarily
based on the original protocol developed by Wang et al.^(46, 47), with slight
modifications. The PCR products were purified using a QIAquick PCR purification
kit (Qiagen) and then sequenced using the Illumina Nova PE150 platform. 2bRAD-M
sequencing was performed at Qingdao OE Biotech Co.,Ltd. (Qingdao, China). Reads
with an N base proportion greater than 8% and low-quality reads (with a base quality
value below Q30 constituting more than 20% of the total bases in a read) were
removed during sequence quality control. To identify the microbial species within
each sample, the sequenced 2bRAD tags underwent quality control and were
subsequently mapped against the 2bRAD marker database using a built-in Perl
script⁽⁴⁷⁾. The relative abundance of a specific species was determined by calculating
the ratio of the number of microbial individuals attributed to that species to the total
number of individuals from known species detectable within a given sample. The
present study employed air samples from laboratory and surgical environments as
negative controls, utilizing a combination of decontam, microDecon, and FEAST to
effectively eliminate background microorganisms from the experimental samples.

The relative abundance feature table was imported into R for further analysis.

The alpha diversity of each sample was evaluated using the Chao 1, Simpson, and
Shannon indices calculated on species level. Compositional differences between each
pair of groups were analyzed using permutational multivariate analysis of variance
(PERMANOVA, 999 permutations), and distance matrix was constructed based on
the Bray–Curtis distance of the relative abundance of species. The compositional shift
was further visualized using principal coordinate analysis (PCoA) based on the same
distance matrix. The alterations at genus, species and KO gene level among different
groups were determined by MaAsLin2 (Microbiome Multivariable Associations with
Linear Models, Maaslin2 R package) with BMI adjusted according to clinical details
of included patients. The significance criteria were prevalence >10% and adjusted q
-value < 0.25 as default⁽²⁵⁾.

*Survival analysis*

Overall survival (OS) includes death from any cause as events after the
perioperative period. For all endpoints, person-time is defined as time from surgery to
event or loss to follow-up (censored). Microbial species associated with OS were
assessed independently in the PDAC group only, using repeated cross-validated
elastic-net penalized Cox proportional hazards regression. Species with relative
abundance under 0.05% and prevalence under 10% were excluded, resulting in the
inclusion of 26 species. We conducted 10-fold cross-validated elastic-net penalized
Cox regression using the “cv.glmnet” function in the “glmnet” R package, with an α
value of 0.5 to allow groups of correlated predictors to be selected together. Clinical

characteristics (age, gender, BMI level, smoking, alcohol, family history, other
malignancy, diabetes, hyperlipemia, antibiotics usage, location, upregulation of
CA19-9, differentiation, perineural invasion (PNI), blood vessel invasion (BVI), T
stage, N stage, M stage, Stage, and neoadjuvant therapy) were included in the model.
The effects of species and clinical characteristics on OS were investigated using
Kaplan–Meier survival curves and compared using log-rank test.

*Metabolome data analysis*

A detailed description of sample preparation, experiment condition, and data
processing for LC-MS and GC-MS untargeted metabolome analysis has been
described in the supplementary materials. In brief, the original LC-MS data were
processed using Progenesis QI V2.3 (Nonlinear, Dynamics, Newcastle, UK) for
baseline filtering, peak identification, integration, retention time correction, peak
alignment, and normalization. The GC/MS raw data were obtained from. The D
format is transferred to. abf format using the software Analysis Base File Converter
for the quick retrieval of data. The data were then imported into MS-DIAL software,
which performs peak detection, peak identification, MS2Dec deconvolution,
characterization, peak alignment, wave filtering, and missing value interpolation.
After the data were normalized, redundancy removal and peak merging were
performed to obtain the data matrix.

The matrix was imported into R for analysis. OPLS-DA was used to identify
differentiating metabolites between the groups. To mitigate overfitting, a 7-fold

cross-validation and 200 Response Permutation Testing (RPT) were performed to
evaluate the model's quality. Variable Importance of Projection (VIP) values derived
from the OPLS-DA model were used to rank the overall contribution of each variable
to group discrimination. Subsequently, a two-tailed Student's t-test was conducted to
verify the statistical significance of the identified metabolites differentiating between
the groups. Differential metabolites were selected based on VIP values greater than
1.0 and p-values less than 0.05. Metabolites origin was analysed using Metorigin.

*Analysis of microbiome-metabolites interactions*

Spearman correlation analysis was performed using the 'psych' package in R to
investigate the associations between differential microbial species, microbial KO
genes and metabolites in PDAC and NAT, respectively. Only associations with a
correlation coefficient (R) greater than 0.3 and an $p < 0.05$ were considered
significant. The resulting associations were visualized using heatmap and network.

*Statistical analyses*

All pairwise comparisons were performed using a two-sided Wilcoxon rank-sum
test (Mann-Whitney U test). Dissimilarity tests among groups (PERMANOVA) were
conducted on Euclidean distance for metabolites and Bray-Curtis distance for
bacteria, with 999 permutations in the R package vegan. Multiple comparisons were
adjusted using Benjamini–Hochberg method. All statistical analyses were performed
using R, version 4.2.1.

***Data availability***

The datasets used and analysed during the current study are included in this
published article and its supplementary information files. No unique code was
generated in this study. The code and raw sequencing data for this study is available
upon request from corresponding author Lei You, florayo@163.com.

**Ethics approval**

Ethical approval for this study was obtained from the Human Research Ethics
Committee of Peking Union Medical College Hospital I-23PJ1417. The requirement
for consent was waived by the ethics committee.

**Acknowledgments**

The authors would like to thank Professor Taiping Zhang for his efforts in
sample collection. Wenbo Kou of OE Biotechnology Company (Qingdao, China) for
his technical support in removing contamination from the sequencing data. This study
was funded by National Multidisciplinary Cooperative Diagnosis and Treatment
Capacity Building Project for Major Diseases, National High Level Hospital Clinical
Research Funding (2022-PUMCH-D-001), Chinese Academy of Medical Sciences
(CAMS) Innovation Fund for Medical Sciences (CIFMS, 2021-I2M-1-002), National
Multidisciplinary Cooperative Diagnosis and Treatment Capacity Building Project for
Major Diseases (NSFC, 81970763), Nonprofit Central Research Institute Fund of
CAMS (PT201832014), National Natural Science Foundation of China (NSFC,

82103016) and (NSFC, 62133006). The funders played no role in study design, data
collection, analysis and interpretation of data, or the writing of this manuscript.

**Conflict of interests**

All authors declare no financial or non-financial competing interests.

**Contributions**

LY and YPZ supervised the study. JR, XW, YF and FHZ collected samples and
provided clinical information. YF performed the bioinformatics analyses. SH have
provided helpful comments. YF and XHL interpreted the results. YF wrote the
manuscript. All authors have read and approved the final manuscript.

**Reference**

- 1. Mizrahi JD, Surana R, Valle JW, Shroff RT. 2020. Pancreatic cancer. *Lancet*
395:2008-2020.
- 2. Rahib L, Wehner MR, Matrisian LM, Nead KT. 2021. Estimated Projection of US
Cancer Incidence and Death to 2040. *JAMA Netw Open* 4:e214708.
- 3. Siegel RL, Miller KD, Wagle NS, Jemal A. 2023. Cancer statistics, 2023. *CA Cancer J*
*Clin* 73:17-48.
- 4. Nevala-Plagemann C, Hidalgo M, Garrido-Laguna I. 2020. From state-of-the-art
treatments to novel therapies for advanced-stage pancreatic cancer. *Nat Rev Clin*

Oncol 17:108-123.

5. Overbeek KA, Levink IJM, Koopmann BDM, Harinck F, Konings I, Ausems M, Wagner
553 A, Fockens P, van Eijck CH, Groot Koerkamp B, Busch ORC, Besselink MG,
Bastiaansen BAJ, van Driel L, Erler NS, Vleggaar FP, Poley JW, Cahen DL, van Hooft
JE, Bruno MJ, Dutch Familial Pancreatic Cancer Surveillance Study G. 2022.
Long-term yield of pancreatic cancer surveillance in high-risk individuals. Gut
71:1152-1160.

6. Hanahan D. 2022. Hallmarks of Cancer: New Dimensions. Cancer Discovery
12:31-46.

7. McQuade JL, Daniel CR, Helmink BA, Wargo JA. 2019. Modulating the microbiome to
improve therapeutic response in cancer. Lancet Oncol 20:e77-e91.

8. Human Microbiome Project C. 2012. Structure, function and diversity of the healthy
human microbiome. Nature 486:207-14.

9. de Martel C, Ferlay J, Franceschi S, Vignat J, Bray F, Forman D, Plummer M. 2012.
Global burden of cancers attributable to infections in 2008: a review and synthetic
analysis. Lancet Oncol 13:607-15.

10. Nejman D, Livyatan I, Fuks G, Gavert N, Zwang Y, Geller LT, Rotter-Maskowitz A,
Weiser R, Mallel G, Gigi E, Meltser A, Douglas GM, Kamer I, Gopalakrishnan V,
Dadosh T, Levin-Zaidman S, Avnet S, Atlan T, Cooper ZA, Arora R, Cogdill AP, Khan
MAW, Ologun G, Bussi Y, Weinberger A, Lotan-Pompan M, Golani O, Perry G, Rokah
571 M, Bahar-Shany K, Rozeman EA, Blank CU, Ronai A, Shaoul R, Amit A, Dorfman T,
Kremer R, Cohen ZR, Harnof S, Siegal T, Yehuda-Shnaidman E, Gal-Yam EN,

Shapira H, Baldini N, Langille MGI, Ben-Nun A, Kaufman B, Nissan A, Golan T,
Dadiani M, et al. 2020. The human tumor microbiome is composed of tumor
type-specific intracellular bacteria. *Science* 368:973-980.

11. Geller LT, Barzily-Rokni M, Danino T, Jonas OH, Shental N, Nejman D, Gavert N,
Zwang Y, Cooper ZA, Shee K, Thaiss CA, Reuben A, Livny J, Avraham R, Frederick
DT, Ligorio M, Chatman K, Johnston SE, Mosher CM, Brandis A, Fuks G, Gurbatri C,
Gopalakrishnan V, Kim M, Hurd MW, Katz M, Fleming J, Maitra A, Smith DA, Skalak
580 M, Bu J, Michaud M, Trauger SA, Barshack I, Golan T, Sandbank J, Flaherty KT,
Mandinova A, Garrett WS, Thayer SP, Ferrone CR, Huttenhower C, Bhatia SN,
Gevers D, Wargo JA, Golub TR, Straussman R. 2017. Potential role of intratumor
bacteria in mediating tumor resistance to the chemotherapeutic drug gemcitabine.
*Science* 357:1156-1160.

12. Thomas RM, Gharaibeh RZ, Gauthier J, Beveridge M, Pope JL, Guijarro MV, Yu Q,
He Z, Ohland C, Newsome R, Trevino J, Hughes SJ, Reinhard M, Winglee K, Fodor
AA, Zajac-Kaye M, Jobin C. 2018. Intestinal microbiota enhances pancreatic
carcinogenesis in preclinical models. *Carcinogenesis* 39:1068-1078.

13. Pushalkar S, Hundeyin M, Daley D, Zambirinis CP, Kurz E, Mishra A, Mohan N, Aykut
B, Usyk M, Torres LE, Werba G, Zhang K, Guo Y, Li Q, Akkad N, Lall S, Wadowski B,
Gutierrez J, Kochen Rossi JA, Herzog JW, Diskin B, Torres-Hernandez A, Leinwand J,
Wang W, Taunk PS, Savadkar S, Janal M, Saxena A, Li X, Cohen D, Sartor RB,
Saxena D, Miller G. 2018. The Pancreatic Cancer Microbiome Promotes Oncogenesis
by Induction of Innate and Adaptive Immune Suppression. *Cancer Discov* 8:403-416.

- 14. Dickson I. 2018. Microbiome promotes pancreatic cancer. *Nat Rev Gastroenterol*
*Hepatol* 15:328.
- 15. Del Castillo E, Meier R, Chung M, Koestler DC, Chen T, Paster BJ, Charpentier KP,
Kelsey KT, Izard J, Michaud DS. 2019. The Microbiomes of Pancreatic and
Duodenum Tissue Overlap and Are Highly Subject Specific but Differ between
Pancreatic Cancer and Noncancer Subjects. *Cancer Epidemiol Biomarkers Prev*
28:370-383.
- 16. Riquelme E, Zhang Y, Zhang L, Montiel M, Zoltan M, Dong W, Quesada P, Sahin I,
Chandra V, San Lucas A, Scheet P, Xu H, Hanash SM, Feng L, Burks JK, Do KA,
Peterson CB, Nejman D, Tzeng CD, Kim MP, Sears CL, Ajami N, Petrosino J, Wood
LD, Maitra A, Straussman R, Katz M, White JR, Jenq R, Wargo J, McAllister F. 2019.

[revised manuscript text omitted]

(E) age. Log-rank test used to generate p-values.

Figure 4. Metabolome profile and function changes in PDAC. (A) and (B) OPLS-DA
showed that PDAC and NAT samples were separated into two distinct clusters. (D, E)
Volcano plot demonstrated metabolites changes between 90 PDAC and 98 NAT
samples. The x-axis indicates log₂-transformed fold change of metabolite
abundances, and the y-axis denotes log₁₀-transformed Q values (p value adjusted
using the tail area-based FDR). The horizontal lines represent $q < 0.05$. (C) and (F)
Venn plot showed results of metabolites origin analysis. (G) The functions of
discriminate metabolites originated from microbiota, metabolic pathways were
studied using the KEGG database, and enriched pathways were displayed by a bubble
plot. The size of bubble represents the number of metabolites detected in the KEGG
pathway.

Figure 5. Integrated analysis of multi-omics in NAT and PDAC. (A, B) The network
revealed representatively significant and suggestive associations ($p < 0.05$, Spearman
analysis) among differentially abundant taxa, metabolites and KO genes (A) NAT
group, and (B) PDAC group. Lines connecting nodes indicate positive (red) or
negative (blue) correlations. (C, D) Representative metabolites-KO gene reactions.
Representative enzymes, metabolites appearing in the existing PDAC metabolite-
enzyme reactions are shown in the formula listed. Each boxplot in a reaction
represents a compound or a KO gene (two-side Wilcoxon rank-sum test). Boxplots

show relative metabolite concentrations or KO gene abundances averaged over
samples within each group and are coloured according to the group. (C) NAT group,
and (D) PDAC group.

Table 1. Characteristics of PDAC patients enrolled.

Characteristics	Count
n(patient)	105
Gender = male (%)	63 (60.0)
Age (mean (SD))	61.57 (8.18)
Family history = yes (%)	20 (19.0)
Pancreatitis = yes (%)	31 (30.1)
Other malignancy = yes (%)	8 (7.6)
BPD = yes (%)	17 (16.2)
EUS FNA = yes (%)	11 (10.5)
Weight loss (mean (SD))	4.19 (4.82)
BMI (mean (SD))	24.76 (3.20)
Smoking (%)	
never	58 (55.2)
ever	12 (11.4)
current	35 (33.3)
Diabetes (%)	
new-onset	14 (13.3)
no	66 (62.9)
yes	25 (23.8)
Alcohol (%)	
never	65 (61.9)
ever	2 (1.9)
current	38 (36.2)
Hyperlipemia (%)	
dyslipidemia	14 (13.3)
no	58 (55.2)
yes	33 (31.4)
Biliary disease = yes (%)	62 (59.0)

Antibiotics = yes (%)	10 (9.5)
Location = tail (%)	44 (41.9)
CA19-9 upregulate = yes (%)	85 (81.0)
Differentiation (%)	
poor	45 (42.9)
moderate	43(41.0)
well	17 (16.1)
PNI = yes (%)	74 (70.5)
BVI = yes (%)	40 (38.1)
T stage (%)	
T1	27 (25.7)
T2	47 (44.8)
T3	28 (26.7)
T4	3 (2.9)
N stage (%)	
N0	51 (48.6)
N1	37 (35.2)
N2	17 (16.2)
M stage = M1 (%)	3 (2.9)
Stage (%)	
IA	15 (14.3)
IB	21 (20.0)
IIA	13 (12.4)
IIB	34 (32.4)
III	19 (18.1)
IV	3 (2.9)
Neoadjuvant therapy = yes (%)	4 (3.8)

Responses to Reviewers

Dear Editors and Reviewers:

Thank you for your letter and for the reviewers' comments concerning our manuscript entitled "Integrated Analysis of Microbiome and Metabolome Reveals Signatures in PDAC Tumorigenesis and Prognosis" (Spectrum00962-24). Those comments are all valuable and very helpful for revising and improving our paper, as well as the important guiding significance to our researches. We appreciate your insights and have taken steps to address the issues raised to improve the quality of the manuscript. We have studied comments carefully and have made correction which we hope meet with approval.

Revised portion are highlighted in yellow in the marked up manuscript. The main corrections in the paper and the responds to the reviewer's comments are as flowing:

Reviewer #1:

1. It is unclear how sequencing data was processed. What are the sequencing depths per sample? How about positive control results? How did you assign taxonomic classification? In the α -diversity comparison, which analysis was used? For α - and β -diversity and taxonomic evaluations, what sequence levels were rarefied per sample? etc. These should be clearly stated in the method section.

Responds to the reviewer's comments: We apologize for the lack of clarity in our methods section. Sequencing depths and quality control results per sample was added into the supplementary Data S1. In the supplementary Materials and Methods, we provide detailed information on taxonomic classification assignment methods. We are very sorry for our negligence of positive control. For α - and β -diversity and taxonomic evaluations, species levels were used, and comparison was analyzed using two-sided Wilcoxon rank-sum test as described in the 'Statistical analyses' section

2. "32 species remained for differential abundance analysis" (page 10, line 175).
"Species with low relative abundance and prevalence were excluded, resulting in the inclusion of 26 species"(page 22, line 441). It is not clear to me why different

taxa remain in the differential analysis. Please list differential abundance analysis and survival analysis for all species in the supplementary Tables.

Responds to the reviewer's comments: The same thresholds were used but different species numbers were obtained, because the analysis of differential abundant microorganisms used PDAC and NAT samples, but survival analysis was performed only in PDAC samples. The relative abundance table of all species detected and species used for survival analysis are supplemented in Data S1. Table S1 lists the results of differential abundance analysis.

3. Controlling for confounding covariates like sex and age, along with known risk factors for PDAC, such as diabetes and smoking status, could also obscure the true relationship between the microbiome and cancer risk. Have you evaluated the association results for the 32 species while factoring in these additional variables? Additionally, have you observed any heterogeneities in the results when conducting subgroup analyses?

Responds to the reviewer's comments:

4. In Fig 3, It is not clear to me why the authors focus on only four pathogens, which were referred to in prostate and breast cancer research. I am also curious about the relative abundance and prevalence of those pathogens in the author's cohort. If the author focuses on opportunistic pathogens, some of the pathogens that have been reported to be associated with pancreatic cancer include *Helicobacter pylori*, *Porphyromonas gingivalis*, *Fusobacterium* species, *Pseudomonas* species, *Staphylococcus* species, *Neisseria elongata* and *Streptococcus mitis*, *Mycobacterium avium*, *Cutibacterium acnes*, etc. could also be considered.

Responds to the reviewer's comments: The findings of adonis analysis (Fig 1C) indicating that the M stage significantly influences the composition of pancreatic microbiota on species level. Subsequently, subgroup analysis was conducted on the M stage to identify distinct bacteria. Figure 3 illustrates the bacteria species associated with the M stage.

5. The authors mention that "A median OS of 36 months was obtained for the

middle-aged group and 17 months for elder patients group ($p = 0.047$) (Figure 3E)." How about the stratification by early onset? Would PDAC patients at younger ages show a different pattern in the impact of *Ralstonia pickettii_B* on OS, separate from the effect of age itself? Conducting survival analysis and also additional controlling for age may change existing results.

Responds to the reviewer's comments: We appreciate the reviewer's thoughtful suggestions and insights. Following reviewer's recommendation, we have conducted a stratified survival analysis and included the results in Supplementary Figure 3. In the corresponding sections of the manuscript, we have described and discussed these findings in detail. However, due to the limited number of early-onset cases, we were unable to create a separate group for this subset in our analysis. Our study's follow-up work is still ongoing, and we anticipate that as we gather more data on patient survival outcomes, the conclusions of our study may evolve.

6. The authors also state that " Microbial species affect the tumorigenesis, metastasis, and prognosis of PDAC via unique microbe-enzyme-metabolite interaction.". How do species affect tumorigenesis, metastasis, and prognosis via interplay analysis? Did you see the different trends or correlations between tumor and normal? What does the MPEA analysis mean?

Responds to the reviewer's comments: We appreciate the valuable suggestion. We have revised our statement to focus solely on the impact of microbial species on tumorigenesis. And based on your recommendations, we have screened and presented the results of the association analysis in NAT and PDAC, respectively. MPEA was abbreviated for metabolite pathway enrichment analysis, the full name of MPEA was added to the article

7. You mentioned, "Differential metabolites were selected based on VIP values greater than 1.0 and p-values less than 0.05." Why not use Multiple comparison adjustments similar to species analysis? However, in the result section, "Differential species analysis between PDAC and NAT groups was conducted using the microbiome multivariable associations with a linear model (MaAsLin 2) and Lefse. Multiple comparisons were adjusted using Benjamini-Hochberg

method, $q < 0.25$." using a q value of 0.25 is indeed higher than the conventional 0.05 and would result in a more lenient threshold, allowing for a greater proportion of false positives in the results.

Responds to the reviewer's comments: In order to effectively analyze metabolomics data, it is necessary to use analysis methods and tools specially designed for the characteristics of metabolomics data, rather than directly applying microbiome analysis methods. Orthogonal Partial Least Squares Discriminant Analysis (OPLS-DA) is a commonly used multivariate statistical method in metabolomics data analysis. By introducing orthogonal signal correction, OPLS-DA enhances model interpretability, reduces unrelated variation, and thereby identifies differential metabolites most relevant to grouping variables. Therefore, we chose to use OPLS-DA instead of applying MaAsLin 2, which was designed specially for data of microbiome, nominal p -values was subjected to multiple hypothesis testing correction using the Benjamini-Hochberg method with an FDR threshold of 0.25 as recommended by the author of MaAsLin 2. The literature of MaAsLin 2 was cited.

8. Based on your statements in Figure 5A and Figure 5B, it appears that you performed a correlation between identified species, KO genes, and metabolites in NAT and PDAC samples, respectively. What metabolites was *Staphylococcus aureus* mostly positively correlated in NAT samples? Could you state the median and range of Spearman correlation for your interest in PDAC and NAT samples, respectively? Does this mean the tumor microbiome metagenome correlation performance with metabolites varies across NAT and PDAC samples?

Responds to the reviewer's comments: Thank you for your detailed observation. We indeed conducted correlation analyses between identified microbial species, KO genes, and metabolites separately in NAT and PDAC samples, we have redrawn figure 5, supplementary figure 5 and supplementary figure 6 based on your suggestion, in order to more clearly demonstrate metabolites that are significantly associated with *Staphylococcus aureus*. In addition, we supplemented the statistically significant association analysis results in supplementary materials Data S1. We apologize for this oversight and ensure to include this information in the revised version.

Reviewer #2:

1. The manuscript was badly written. Many abbreviations are used before spelling out. The digits of numbers are not consistent. Many microbial taxa names are not italicized. Unless comprehensive English editing was performed by a local writer, the writing quality of this work does not align with the average levels of paper published in mSystems.

Responds to the reviewer's comments: We have extensively revised the manuscript to ensure smooth flow of writing, consistency in terminology, rectified the issue of using abbreviations without explanation throughout the manuscript and ensured consistency in the presentation of numerical data. All microbial taxa names at genus and species level have been italicized as per the requirements and standards of the journal.

2. I'm quite concerned about the soundness of the microbiome profiling technology used in this study. The original paper describing this technology was published more than 20 years ago and has been rarely used in microbiome studies in the past 5 years. Also, the authors described that they identified a total of 1,920 species but only 32 of them retained for differential abundance analyses after filtering low prevalence and abundance taxa. If this is not a typo, it means that >98% of species are of low prevalence or abundance. I do not think a technology is sound if >98% signals it identifies are of low quality. Unless this concern can be fully addressed, I can not endorse the publication of this work.

Responds to the reviewer's comments: The microbiome profiling technology used in this study was 2bRAD-M, we are sorry for only referencing the methodology literature of 2bRAD-M in the supplementary methods previously. The literature has now been cited in the revised manuscript. The publication is as follows: Sun, Z., Huang, S., Zhu, P. et al. Species-resolved sequencing of low-biomass or degraded microbiomes using 2bRAD-M. *Genome Biol* 23, 36 (2022). <https://doi.org/10.1186/s13059-021-02576-9>. 2bRAD-M is a highly reduced and cost-effective strategy which only sequences ~1% of metagenome and can simultaneously produce species-level bacterial, archaeal, and fungal profiles. 2bRAD-M can accurately generate species-level taxonomic profiles for otherwise

hard-to-sequence samples with merely 1 pg of total DNA, high host DNA contamination, or severely fragmented DNA from degraded samples.

3. The analysis of genus-level taxa does not make sense to me because it was derived using only 32 species. This derived genus-level information deviated a lot from the real genus-level information.

Responds to the reviewer's comments: We re-analyzed the genus-level bacteria using the MaAsLin 2 default parameters, i.e., prevalence > 0.1, $q < 0.25$, and the results were consistent with the previous ones, and the description of the analysis has been supplemented in the Methods section.

4. "The 179 model included group and M stage as fixed effects and BMI as a random effect." - I do not think this is the correct usage of MaAsLin 2. Fixed and random effect variables are incorrectly assigned.

Responds to the reviewer's comments: We used group and M stage as fixed effects and BMI as a random effect, as well as group as fixed effects and BMI, M stage as random effects, and got the same differential species for PDAC group.

5. It is not clear how PICRUSt2 was used to infer functional features. If the authors use the 32 species, it is hard to believe 6,083 KO genes could be identified. While, if the 1,920 species was used, it does not make sense because >98% of these species are of low prevalence/abundance.

Responds to the reviewer's comments: The 1,920 species was used for PICRUSt2, and a total of 7301 KO genes was detected, the '6,083 KO genes' was a result after filtering out low abundant and low prevalent KO genes. We re-analyzed the KO-genes using the MaAsLin 2 default parameters, i.e., prevalence > 0.1, $q < 0.25$, and the results were consistent with the previous ones, and the description of the analysis has been supplemented in the Methods section.

6. Negative controls were used in DNA extraction and PCR amplification process. However, it is not clear whether microbial profiling was performed for negative control samples. *Cutibacterium acnes* stands out as one of the differentially abundance species. This species is more commonly seen in the skin microbiota and linked to acne. This species might be derived from contamination.

Responds to the reviewer's comments: Regarding your concern about microbial profiling of negative control samples, we apologize for not clearly stating in the manuscript that microbial profiling was indeed conducted for not only in DNA extraction and PCR amplification process, but also in environment samples of operating room and 1 laboratory. Microbial profiling was performed for negative control samples, and using a combination of decontam, microDecon, and FEAST, background microorganisms in the samples were deducted based on the negative controls.

7. It is not clear whether the patients underwent treatment and the type of treatment. Treatment should be an important covariant that should be adjusted in all analyses.

Responds to the reviewer's comments: Thank you for your feedback regarding the treatment information of patients in our study. We appreciate your attention to this aspect of our research. We collected information on preoperative antibiotic treatment, biliary stent placement, and neoadjuvant therapy for all patients. Our Adonis analysis revealed that these treatments did not significantly affect the microbial composition of pancreatic tissue. Regarding postoperative chemotherapy information, our follow-up is ongoing and we are working to collect and update these data. We will include your recommendations for adjusting treatment as an important covariate in our analyses in follow-up studies.

Sincerely,

Corresponding authors:

Lei You, E-mail: florayo@163.com

Yupei Zhao, E-mail: zhao8028@263.net

Responses to Reviewers

Dear Editors and Reviewers:

Thank you for your letter and for the reviewers' comments concerning our manuscript entitled "Integrated Analysis of Microbiome and Metabolome Reveals Signatures in PDAC Tumorigenesis and Prognosis" (Spectrum00962-24). Those comments are all valuable and very helpful for revising and improving our paper, as well as the important guiding significance to our researches. We appreciate your insights and have taken steps to address the issues raised to improve the quality of the manuscript. We have studied comments carefully and have made correction which we hope meet with approval.

Revised portion are highlighted in yellow in the marked up manuscript. The main corrections in the paper and the responds to the reviewer's comments are as flowing:

Reviewer #1:

1. Figure 3A - Presentation of Incidental Results:

It is unusual to include incidental findings in the main figures of a manuscript as this might distract from the main messages of the study. Consider moving these results to supplementary materials unless they are directly relevant to the main research questions addressed in your paper.

Responds to the reviewer's comments: Thank you for your suggestion. In this revision, we have included figures of the species related to the tumor M stage (Figure 3A-C) in the supplementary materials.

2. Methods Section - Survival Analysis:

Please specify how many times the 10-fold cross-validation was repeated in your survival analysis. The selection of *Ralstonia pickettii_B* needs clearer justification. Out of the 26 microbial species and clinical characteristics evaluated, Detail the selection criteria and the statistical thresholds used for including this species in your model.

Responds to the reviewer's comments: Thank you for bringing up the issue of our

vague description. In the Methods Section - Survival Analysis of this revision, we have provided a more detailed description of the thresholds for bacterial species included in the survival analysis, the parameters used in constructing the regularized Cox model (specifically the values of alpha and lambda), and the workflow for the survival analysis. (1) Microbial species with relative abundance under 0.05% and prevalence under 10% were excluded. (2) We built an elastic-net penalized Cox regression model using the “glmnet” function in the “glmnet” R package, with an α value of 0.5 to allow groups of correlated predictors to be selected. 10-fold cross-validation was conducted once using the “cv.glmnet” function to determine the value of optimal lambda.1se to build a regularized Cox model with the fewest variables.

3. Page 10, Line 194 - Kaplan-Meier Curve and Survival Analysis:

You mention the use of Kaplan-Meier curves tested by the log-rank test. While this is a valid approach for univariate survival analysis, the relationship and comparison with Cox proportional hazards regression results should also be clarified. Please include the results of the standard Cox regression and the outputs from the repeated 10-fold cross-validation for the 26 species.

Responds to the reviewer’s comments: Thank you for your advice. We apologize for the lack of clarity in the previous survival analysis workflow. The Cox regression model we built was an elastic-net penalized Cox regression model. For which, the 10-fold cross-validation was conducted once to determine the value of optimal lambda.1se. In this revision, we conducted a standard univariate Cox regression analysis for 26 species included in the survival analysis. We added the standard univariate Cox regression outputs to the supplementary materials in table form and the hazard ratio of *Ralstonia pickettii*_B and age in Results Section -Microbial species related to overall survival.

4. Please thoroughly revise the manuscript to correct all grammatical errors and enhance the clarity of arguments. Including "For α -diversity, significant decrease was observed in the in PDAC" in line 141; "On the one hand, if this is a limit, the comparison of matched tumor and NAT areas minimized... " on page 15, line 296.

et al.

Responds to the reviewer's comments: Thanks to your suggestion. We have revised the article's grammar and clarified some of the language to improve expression.

Reviewer #2:

1. The manuscript was badly written. Many abbreviations are used before spelling out. The digits of numbers are not consistent. Many microbial taxa names are not italicized. Unless comprehensive English editing was performed by a local writer, the writing quality of this work does not align with the average levels of paper published in mSystems.

Responds to the reviewer's comments: We have extensively revised the manuscript to ensure a smooth flow of writing and consistency in terminology, rectified the issue of using abbreviations without explanation throughout the manuscript, and ensured consistency in the presentation of numerical data. All microbial taxa names at the genus and species level have been italicized as per the requirements and standards of the journal.

2. I'm quite concerned about the soundness of the microbiome profiling technology used in this study. The original paper describing this technology was published more than 20 years ago and has been rarely used in microbiome studies in the past 5 years. Also, the authors described that they identified a total of 1,920 species but only 32 of them retained for differential abundance analyses after filtering low prevalence and abundance taxa. If this is not a typo, it means that >98% of species are of low prevalence or abundance. I do not think a technology is sound if >98% signals it identifies are of low quality. Unless this concern can be fully addressed, I can not endorse the publication of this work.

Responds to the reviewer's comments: The microbiome profiling technology used in this study was 2bRAD-M, we are sorry for only referencing the methodology literature of 2bRAD-M in the supplementary methods previously. The literature has now been cited in the revised manuscript. The publication is as follows: Sun, Z., Huang, S., Zhu, P. et al. Species-resolved sequencing of low-biomass or degraded microbiomes using 2bRAD-M. *Genome Biol* 23, 36 (2022).

<https://doi.org/10.1186/s13059-021-02576-9>. 2bRAD-M is a highly reduced and cost-effective strategy that only sequences ~1% of metagenome and can simultaneously produce species-level bacterial, archaeal, and fungal profiles. 2bRAD-M can accurately generate species-level taxonomic profiles for otherwise hard-to-sequence samples with merely one pg of total DNA, high host DNA contamination, or severely fragmented DNA from degraded samples.

3. The analysis of genus-level taxa does not make sense to me because it was derived using only 32 species. This derived genus-level information deviated a lot from the real genus-level information.

Responds to the reviewer's comments: We re-analyzed the genus-level bacteria using the MaAsLin 2 default parameters, i.e., prevalence > 0.1, $q < 0.25$, and the results were consistent with the previous ones, and the description of the analysis has been supplemented in the Methods section.

4. "The 179 model included group and M stage as fixed effects and BMI as a random effect." - I do not think this is the correct usage of MaAsLin 2. Fixed and random effect variables are incorrectly assigned.

Responds to the reviewer's comments: We used group and M stage as fixed effects and BMI as a random effect, as well as group as fixed effects and BMI, M stage as random effects, and got the same differential species for PDAC group.

5. It is not clear how PICRUSt2 was used to infer functional features. If the authors use the 32 species, it is hard to believe 6,083 KO genes could be identified. While, if the 1,920 species was used, it does not make sense because >98% of these species are of low prevalence/abundance.

Responds to the reviewer's comments: The 1,920 species was used for PICRUSt2, and a total of 7301 KO genes was detected, the '6,083 KO genes' was a result after filtering out low abundant and low prevalent KO genes. We re-analyzed the KO-genes using the MaAsLin 2 default parameters, i.e., prevalence > 0.1, $q < 0.25$, and the results were consistent with the previous ones, and the description of the analysis has been supplemented in the Methods section.

6. Negative controls were used in DNA extraction and PCR amplification process.

However, it is not clear whether microbial profiling was performed for negative control samples. *Cutibacterium acnes* stands out as one of the differentially abundance species. This species is more commonly seen in the skin microbiota and linked to acne. This species might be derived from contamination.

Responds to the reviewer's comments: Regarding your concern about microbial profiling of negative control samples, we apologize for not clearly stating in the manuscript that microbial profiling was indeed conducted for not only in DNA extraction and PCR amplification process, but also in environment samples of operating room and laboratory. Microbial profiling was performed for negative control samples, and using a combination of decontam, microDecon, and FEAST, background microorganisms in the samples were deducted based on the negative controls.

7. It is not clear whether the patients underwent treatment and the type of treatment. Treatment should be an important covariant that should be adjusted in all analyses.

Responds to the reviewer's comments: Thank you for your feedback regarding the treatment information of patients in our study. We appreciate your attention to this aspect of our research. We collected information on preoperative antibiotic treatment, biliary stent placement, and neoadjuvant therapy for all patients. Our PERMANOVA analysis revealed that these treatments did not significantly affect the microbial composition of pancreatic tissue. Regarding postoperative chemotherapy information, our follow-up is ongoing and we are working to collect and update these data. We will include your recommendations for adjusting treatment as an important covariate in our analyses in follow-up studies.

Sincerely,

Corresponding authors:

Lei You, E-mail: florayo@163.com

Yupei Zhao, E-mail: zhao8028@263.net

Re: Spectrum00962-24R1 (Integrated Analysis of Microbiome and Metabolome Reveals Signatures in PDAC Tumorigenesis and Prognosis)

Dear Dr. Lei You:

Thank you for the privilege of reviewing your work. Below you will find my comments, instructions from the Spectrum editorial office, and the reviewer comments.

As pointed out by the reviewers, there are a lot of issues ranging from research methodology (lack of some details, possible contamination, reproducibility...), results (additional data and validations are required to support the conclusion), discussion (limitation and novelty of current study needs to be carefully discussed). Instead of declining this ms for publications, I would offer authors a chance to consider if they can address the concerns with additional data in next 2-3 months.

Revision Guidelines

Sincerely,
Pei-Yuan Qian
Editor
Microbiology Spectrum

Reviewer #1 (Comments for the Author):

In this manuscript, Fang et al. conducted a large sample-size pancreatic adenocarcinoma microbiome study using a novel

microbiome sequencing method and two metabolomic assays. They identified differences in overall microbial diversity and composition, opportunist pathogenic species, and metabolites between PDAC and adjacent normal tissues, one of which is associated with worse OS.

The authors do make an effort to address my comments, and I feel that the quality of the manuscript has improved. Some of my previous comments have been addressed. However, there are still some areas needing further attention:

1 Validation of Results: A one-time 10-fold cross-validation (CV) might not be sufficient. I recommend repeating this process 10 to 20 times to ensure the reliability of the performance metrics or species rankings. The averaged results could provide a more robust basis for conclusions.

2 Survival Analysis: The univariate Cox proportional hazards (PH) model could be enhanced by including a multivariate model that adjusts for potential confounders such as stage, cigarette smoking, and BMI. There is a discrepancy in the p-values reported for *Ralstonia pickettii_B* between Table S2 ($p=0.055641$) and Figure 3 ($p=0.045$), which raises concerns about the data coding and processing. It may be more informative to analyze the relative abundance of microbial species (clr-transformed etc) in relation to OS, rather than just their presence.

3 Technical Corrections: Please correct the abbreviation 'lefse' (line 720) to LEfSe. Case-insensitivity, incorrect abbreviations, non-italicized microbial species names, and confusing data presentation still occur in Figure legends, figures, tables, and supplementals; these significantly detract from the clarity and professionalism of the manuscript. Additionally, ensure that the references are updated to reflect the most recent statistics, such as the 5-year overall survival rate of 13% for PDAC in 2024, etc.

Reviewer #5 (Comments for the Author):

Fang et al. conducted a comparative analysis of microbial profiles and their associated metabolites in cancerous and paraneoplastic tissues from 105 patients with pancreatic ductal adenocarcinoma (PDAC) using microbiome and metabolome sequencing. The study revealed significant differences in microbial and metabolic molecules between PDAC cancerous and paracancerous tissues, with key microbes associated with patient's prognosis. These findings suggest that intratumoral microbes may influence PDAC occurrence and development through specific metabolites. Although this multi-omics study has expanded the understanding of intratumoral microbes in PDAC, several critical issues need to be addressed:

Major Issues:

1. The abundance of intratumoral microbes is very low, and contamination during specimen collection is a significant concern. Preventing contamination is crucial, and the study should adhere to the RIDE checklist for contamination control. The study, spanning six years, is retrospective, increasing the likelihood of contamination. Although the authors implemented several controls, they did not provide specific descriptions of these controls, details of their collection, the contaminating bacteria, or the principles of their treatment. The presence of many microorganisms that are typically found in the environment but are rare or absent in humans suggests a high probability of contamination. The authors need to provide tables of abundance for experimental and control groups for verification.
2. It is unclear how the study determined that the detected metabolites originated from intratumoral microorganisms, as relying solely on Spearman correlation analysis is insufficient.
3. The study lacks external validation, and the strain composition and diversity found are inconsistent with previous reports, reducing the study's reliability.
4. The study's conclusions appear to be exaggerated, making it impossible to draw definitive conclusions from the current findings.

Minor Issues:

1. It is recommended to supplement the study with metabolome QC maps.
2. A comparative analysis of microorganisms according to the length of time of collection is suggested.
3. The criteria for determining the correlation threshold should be clarified.
4. The criteria and thresholds for selecting intratumoral microbes, given that 98% of detected sequences or material are rejected, need to be specified.

Review Comments for Manuscript ID [Spectrum00962-24R1]

General Comments:

In this manuscript, Fang et al. conducted a large sample-size pancreatic adenocarcinoma microbiome study using a novel microbiome sequencing method and two metabolomic assays. They identified differences in overall microbial diversity and composition, opportunist pathogenic species, and metabolites between PDAC and adjacent normal tissues, one of which is associated with worse OS.

The authors do make an effort to address my comments, and I feel that the quality of the manuscript has improved. Some of my previous comments have been addressed. However, there are still some areas needing further attention:

Major Comments:

1 Validation of Results: A one-time 10-fold cross-validation (CV) might not be sufficient. I recommend repeating this process 10 to 20 times to ensure the reliability of the performance metrics or species rankings. The averaged results could provide a more robust basis for conclusions.

2 Survival Analysis: The univariate Cox proportional hazards (PH) model could be enhanced by including a multivariate model that adjusts for potential confounders such as stage, cigarette smoking, and BMI. There is a discrepancy in the p-values reported for *Ralstonia pickettii_B* between Table S2 (p=0.055641) and Figure 3 (p=0.045), which raises concerns about the data coding and processing. It may be more informative to analyze the relative abundance of microbial species (clr-transformed etc) in relation to OS, rather than just their presence.

Minor Comments:

3 Technical Corrections: Please correct the abbreviation 'lefse' (line 720) to LEfSe. Case-insensitivity, incorrect abbreviations, non-italicized microbial species names, and confusing data presentation still occur in Figure legends, figures, tables, and supplementals; these significantly detract from the clarity and professionalism of the manuscript. Additionally, ensure that the references are updated to reflect the most recent statistics, such as the 5-year overall survival rate of 13% for PDAC in 2024, etc.

Response letter

Dear Editors and Reviewers:

Thank you for your letter and for the reviewers' comments concerning our manuscript entitled "Integrated Analysis of Microbiome and Metabolome Reveals Signatures in PDAC Tumorigenesis and Prognosis" (Spectrum00962-24R1). Those comments are all valuable and very helpful for revising and improving our paper, as well as the important guiding significance to our researches. We appreciate your insights and have taken steps to address the issues raised to improve the quality of the manuscript. We have studied comments carefully and have made correction which we hope meet with approval.

Revised portion are marked in red in the marked-up manuscript. The main corrections in the paper and the responds to the reviewer's comments are as flowing:

Reviewer #1:

1.Validation of Results: A one-time 10-fold cross-validation (CV) might not be sufficient. I recommend repeating this process 10 to 20 times to ensure the reliability of the performance metrics or species rankings. The averaged results could provide a more robust basis for conclusions.

Responds to the reviewer's comments: Thank you for your valuable suggestion. We greatly appreciate your point regarding the importance of cross-validation. In this revision, we conducted 100 times 10-fold cross-validation to ensure the reliability of the performance metrics and species rankings. Same result was obtained after revision. The 'Survival analysis' part of the Materials and methods section and the 'Microbial species related to overall survival' part of Results section has been modified

2.Survival Analysis: The univariate Cox proportional hazards (PH) model could be enhanced by including a multivariate model that adjusts for potential confounders such as stage, cigarette smoking, and BMI. There is a discrepancy in the p-values reported for *Ralstonia pickettii_B* between Table S2 ($p=0.055641$) and Figure 3 ($p=0.045$), which raises concerns about the data coding and processing. It may be more informative to analyze the relative abundance of

microbial species (clr-transformed etc) in relation to OS, rather than just their presence.

Responds to the reviewer's comments: Thank you very much for your insightful feedback. The method you suggested is indeed a common approach in survival analysis.

(1) In the early stages of our study, we utilized the analysis methods and concepts you recommended. We initially performed univariate Cox regression using the relative abundance and clr-transformed relative abundance of the bacteria species, followed by multivariate Cox regression based on the results from the univariate analysis. However, we encountered particularly large confidence intervals for the hazard ratios obtained from this method. Upon reviewing the literature, we identified two key reasons for this phenomenon: first, microbiome data is zero-inflated, and this data structure can result in unreliable hazard ratio outcomes when using standardized Cox regression; second, our sample size was limited, a larger sample size can improve this outcome. We plan to address this issue by increasing our sample size in future research. Therefore, in this study, we analyzed bacterial presence when performing univariate COX regression analysis

(2) Building on the results of our preliminary analysis, we referenced published literature and adopted an elastic net regularization approach combined with Cox regression for our analysis. This method allowed us to identify factors that influence the overall survival of pancreatic cancer patients based on the relative abundance of bacterial species and metadata, and in line with your suggestion, we conducted 100 times 10-fold cross-validation to ensure the robustness of our findings. We summed the number of times each factor was selected out of the 100 repetitions. We focused further on factors selected $\geq 50\%$ of the 100 times (50 times or more) and with $p < 0.20$ in standard univariate Cox proportional hazards models. Count of times which factors were selected in the repetition was added in Table S2. The reference literatures are as follows:

Peters BA, Wilson M, Moran U, et al. Relating the gut metagenome and metatranscriptome to immunotherapy responses in melanoma patients. *Genome Med.*

2019;11(1):61. Published 2019 Oct 9. doi:10.1186/s13073-019-0672-4.

Peters BA, Pass HI, Burk RD, et al. The lung microbiome, peripheral gene expression, and recurrence-free survival after resection of stage II non-small cell lung cancer. *Genome Med.* 2022;14(1):121. Published 2022 Oct 27. doi:10.1186/s13073-022-01126-7.

(3) The inconsistency in the p-values presented in Table S2 (p=0.055641) and Figure 3 (p=0.045) arises because the p-values in the Figure 3 correspond to the log-rank test results from the Kaplan-Meier survival curves, while the p-values in the Table S2 are derived from the univariate Cox regression analysis.

Thank you once again for your constructive comments, and we hope that these clarifications address your concerns.

3. Technical Corrections: Please correct the abbreviation 'lefse' (line 720) to LEfSe.

Case-insensitivity, incorrect abbreviations, non-italicized microbial species names, and confusing data presentation still occur in Figure legends, figures, tables, and supplementals; these significantly detract from the clarity and professionalism of the manuscript. Additionally, ensure that the references are updated to reflect the most recent statistics, such as the 5-year overall survival rate of 13% for PDAC in 2024, etc.

Responds to the reviewer's comments: Thank you for your advice. We appreciate your detailed suggestions, which helped enhance the quality and clarity of our work. Firstly, we corrected the abbreviation "lefse" to "LEfSe" as you noted. Additionally, we conducted a thorough review of Figure legends, figures, tables, and supplementals to ensure consistency and accuracy, making the necessary corrections to enhance readability and professionalism. Lastly, we updated our references, incorporating the most recent statistics as you suggested. In the Discussion section, we have also updated the comparison with other studies of intratumoral microbiota in pancreatic cancer and the corresponding literature. Thank you once again for your constructive comments. We look forward to making these revisions and improving our manuscript accordingly.

Reviewer #2:

Major Issues:

1. The abundance of intratumoral microbes is very low, and contamination during specimen collection is a significant concern. Preventing contamination is crucial, and the study should adhere to the RIDE checklist for contamination control. The study, spanning six years, is retrospective, increasing the likelihood of contamination. Although the authors implemented several controls, they did not provide specific descriptions of these controls, details of their collection, the contaminating bacteria, or the principles of their treatment. The presence of many microorganisms that are typically found in the environment but are rare or absent in humans suggests a high probability of contamination. The authors need to provide tables of abundance for experimental and control groups for verification.

Responds to the reviewer's comments: Thank you for your thorough review and your constructive comments regarding the potential contamination issues in our study. We appreciate your insights, as they highlight critical aspects that need to be addressed to enhance the rigor of our research.

(1) To address your comments, we provided a detailed description of the specific controls we implemented to minimize contamination risk in 'Microbiota analysis' part in the Materials and methods section as follows:

The present study employed air samples from laboratory and surgical environments as negative controls, utilizing a combination of decontam, microDecon, and FEAST to eliminate background microorganisms from the experimental samples effectively. Contaminants were identified based on tissue samples and environmental control samples using microDecon's decon function and decontam's isContaminant function. Based on the list of contaminants identified by microDecon and decontam, the union set is taken as the final pollutant list. The proportion of unknown origin calculated using FEAST replaces the value of the contaminant in the experimental group sample. Each experimental sample was normalized by dividing by the sum of the samples to obtain the relative abundance after decontamination. Decontamination analyses used default parameters.

(2) Additionally, the relative abundance table of species in environmental control samples was included in the Supplementary dataset.

Thank you once again for your invaluable feedback. We are committed to improving the clarity and robustness of our manuscript in response to your concerns.

2. It is unclear how the study determined that the detected metabolites originated from intratumoral microorganisms, as relying solely on Spearman correlation analysis is insufficient.

Responds to the reviewer's comments: Thank you for your valuable feedback regarding the determination of metabolite origins in our study. We appreciate your concern about the methods used to establish the connection between detected metabolites and intratumoral microorganisms. In addition to the Spearman correlation analysis, we employed MetOrigin to conduct a comprehensive analysis of the sources of the metabolites. MetOrigin is designed to assess the potential origins of metabolites, the determination of the origins of metabolites currently depends on the integration of seven powerful databases, including HMDB, BIGG, ChEBI, FoodDB, Drugbank, and T3DB. So far, MetOrigin has included a total of 191,031 metabolites that contain specific sources, including host (mammals), microbiota (archaea, fungi, bacteria), cometabolism (shared by both host and microbiota), or from food (food & plant), drug, and environment (toxins & pollutants). By integrating this metabolic source analysis, we strengthened our ability to attribute the detected metabolites directly to the identified intratumoral microbes. The reference literature of MetOrigin is as follows:

Yu, Gang, Cuifang Xu, Danni Zhang, Feng Ju, and Yan Ni. 2022. MetOrigin: Discriminating the Origins of Microbial Metabolites for Integrative Analysis of the Gut Microbiome and Metabolome. *iMeta*. 1, e10. <https://doi.org/10.1002/imt2.10>

3. The study lacks external validation, and the strain composition and diversity found are inconsistent with previous reports, reducing the study's reliability.

Responds to the reviewer's comments: Thank you for your thoughtful feedback regarding the need for external validation and the concerns about the strain composition and diversity findings in our study. We appreciate your insights, as they are crucial for enhancing the quality and reliability of our research. We acknowledge

that external validation is an important aspect of any study, particularly in microbiome research. While our current study focuses on a specific patient cohort, we recognize the necessity of validating our findings with additional external datasets or independent cohorts in future research. Regarding the inconsistencies in strain composition and diversity compared to previous reports, we appreciate your observations. We have investigated these discrepancies further and provide a more comprehensive discussion in the revised manuscript. Thank you again for your valuable comments. Your feedback is instrumental in guiding us to improve our manuscript and the overall robustness of our findings.

4.The study's conclusions appear to be exaggerated, making it impossible to draw definitive conclusions from the current findings.

Responds to the reviewer's comments: Thank you for your insightful feedback regarding the conclusions drawn in our study. We appreciate your perspective on the need for more measured language in our conclusions. In response to your comments, we have carefully revised the wording of the conclusion section to better reflect the findings and avoid any exaggeration. We have taken great care to ensure that our conclusions are substantively aligned with the data and evidence presented in the study, emphasizing the limitations and the context of our findings.

Minor Issues:

1.It is recommended to supplement the study with metabolome QC maps.

Responds to the reviewer's comments: Thank you for your valuable recommendation. In response to your suggestion, we supplemented our manuscript with metabolome QC maps as Supplementary figure 7 to enhance the robustness and reliability of our findings.

2.A comparative analysis of microorganisms according to the length of time of collection is suggested.

Responds to the reviewer's comments: Thank you for your thoughtful suggestion. Given the low prevalence of pancreatic cancer, it is, unfortunately, inevitable that our research relies on a longer time span for sample collection to achieve a sufficiently large sample size for microbiome studies. However, we want to assure you that we

have utilized 2bRAD-M sequencing which is effective in analyzing low-biomass, high-host-contaminated, and degraded samples, thereby ensuring the reliability of our findings. To further address concerns about sample integrity, we would like to clarify that tissue samples were collected during surgery and immediately placed into sterile tubes. They were subsequently stored at -80°C for microbiome sequencing. Throughout the collection process, we took extensive precautions to maintain sterility and minimize any risk of contamination. While we understand the value of a comparative analysis based on collection time, we believe that conducting such an analysis may introduce complexities that could confound our results and detract from the primary objectives of our study. Therefore, we respectfully choose not to pursue this specific comparative analysis at this time. Thank you once again for your valuable feedback, which has helped us consider important aspects of our research design.

3.The criteria for determining the correlation threshold should be clarified.

Responds to the reviewer's comments: Thank you for your insightful comment. The criteria we utilized for setting the correlation threshold in our analysis was an absolute correlation coefficient (R) value greater than 0.3 and $p < 0.05$, which was based on both statistical significance and biological relevance. In this revision, we conducted a comprehensive review of all sections of the manuscript, supplemental materials, and figure legends related to correlation analysis. We have ensured that the correlation threshold is explicitly added. Thank you once again for your valuable feedback, which has greatly contributed to improving our manuscript.

4.The criteria and thresholds for selecting intratumoral microbes, given that 98% of detected sequences or material are rejected, need to be specified.

Responds to the reviewer's comments: Thank you for your important feedback regarding the criteria and thresholds used for selecting intratumoral microbes. To clarify, the criteria we employed for selecting intratumoral microbes were specifically defined as follows: we included species with a total relative abundance greater than 0.05% and a prevalence of greater than 10%. This approach allowed us to focus on

the microbes that, while present at low abundance, are consistently found across samples, thereby enhancing the relevance of our findings. These criteria were specified in the revised manuscript in the 'Microbiota analysis' part in Materials and methods section. Thank you once again for your valuable suggestions, which contribute significantly to our manuscript's clarity and rigor.

Sincerely,

Corresponding authors:

Lei You, E-mail: florayo@163.com

Yupei Zhao, E-mail: zhao8028@263.net

Re: Spectrum00962-24R2 (Integrated Analysis of Microbiome and Metabolome Reveals Signatures in PDAC Tumorigenesis and Prognosis)

Dear Dr. Lei You:

The ms has been improved substantially via review process and the revised version is ready to be published.

Your manuscript has been accepted, and I am forwarding it to the ASM production staff for publication. Your paper will first be checked to make sure all elements meet the technical requirements. ASM staff will contact you if anything needs to be revised before copyediting and production can begin. Otherwise, you will be notified when your proofs are ready to be viewed.

Sincerely,
Pei-Yuan Qian
Editor
Microbiology Spectrum

Reviewer #1 (Comments for the Author):

I believe the author has adequately addressed my concerns and doubts regarding the methodology, and the quality of the paper has significantly improved. It is suitable for publication in this journal.

Reviewer #4 (Comments for the Author):

The authors have addressed all of my concerns.